# AN EFFICIENT RUBRIC-BASED GENERATIVE VERIFIER FOR SEARCH-AUGMENTED LLMS

## ABSTRACT

Search augmentation empowers Large Language Models with retrieval capabilities to overcome the limitations imposed by static parameters. Recently, Reinforcement Learning leverages tailored reward signals as a viable technique to enhance LLMs performing tasks involving search. However, existing reward modeling for search-augmented LLMs faces several limitations. Rule-based rewards, such as Exact Match, are verifiable but fragile to variations in expression and cannot be applied to long-form workloads. In contrast, generative rewards improve robustness, but designing verifiable and stable rewards for long-form workloads in dynamic corpora remains challenging and also incurs high computational costs. In this paper, we propose a unified and verifiable paradigm, "nugget-as-rubric", which treats atomic information points as structured evaluation criteria for different search-augmentation workloads. Short-form tasks correspond to a single rubric, whereas long-form tasks expand to multiple rubrics aligned with the question's information needs. To support long-form settings, we design an automatic rubric construction pipeline based on query rewriting, which can automatically retrieve passages relevant to each question and extract rubrics from them, both from static corpora and from dynamic online web content. Furthermore, we introduce **Search-Gen-V**, a 4B-parameter efficient generative verifier under our proposed verifiable paradigm, which is trained via the idea of distillation and a two-stage strategy. Experimental results show that Search-Gen-V achieves strong verification accuracy across different workloads, making it a scalable, robust, and efficient verifiable reward constructor for search-augmented LLMs. [1]

## 1 INTRODUCTION

Search augmentation (Lewis et al., 2021; Gao et al., 2024) refers to endowing Large Language Models (LLMs; Zhao et al., 2025; Brown et al., 2020) with search capabilities for reasoning and generation, thereby overcoming the limitations imposed by static parameters (Zhang et al., 2023; Huang et al., 2025a). Under this paradigm, relevant and up-to-date external information can be recalled on demand to support LLMs in performing factual tasks, or formed into a retrieval environment where agentic LLMs can autonomously conduct multi-turn information retrieval (Li et al., 2025a; Xi et al., 2025). In this line of research, a key challenge is discovering effective techniques to stimulate models to exhibit stronger search capabilities. Prompt-based approaches frequently suffer from limited generalization (Trivedi et al., 2023), whereas supervised fine-tuning (SFT) not only depends heavily on the availability of large-scale, high-quality annotated trajectories but also risks trapping models in a "memorization" pitfall (Schick et al., 2023; Chu et al., 2025). More recently, notable breakthroughs have been reported with methods based on Reinforcement Learning (RL; Jin et al., 2025; Song et al., 2025; Gao et al., 2025). This can be largely attributed to the well-designed reward signals, which provide feedback to refine the model's search behavior. Thus, reward modeling is crucial for further improving search-augmented LLMs.

The design of rewards is closely tied to the objectives of search augmentation. At present, search-augmented LLMs primarily confront two types of workloads:

---

[1]**Code:** https://anonymous.4open.science/r/ICLR-Rebuttal-C82D

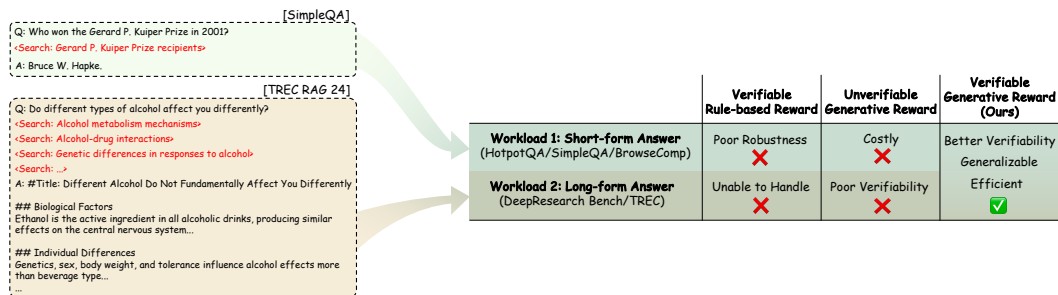

Figure 1: Two typical workloads for search-augmented LLMs. Existing reward modeling methods suffer from issues in robustness, verifiability, and computational cost. Our approach in this work not only unifies both types of workloads but also achieves better verifiability and higher efficiency.

- *Short-form answer*, typically involves only a single information point (consisting of a specific entity name). Representative datasets include HotpotQA (Yang et al., 2018), SimpleQA (Wei et al., 2024), and BrowseComp (Wei et al., 2025).
- *Long-form answer*, requires multiple information points and can usually reach the paragraph level or report level. Representative datasets include DeepResearch Bench (Du et al., 2025) and TREC datasets (Craswell et al., 2025a;b;c).

Rule-based reward models, which rely on scoring functions such as Exact Match and F1 Score, are commonly utilized for short-form workloads, and fall under the paradigm of Reinforcement Learning with Verifiable Rewards (RLVR; Lambert et al., 2025; DeepSeek-AI et al., 2025). While verifiable, they lack robustness to variations in expression (e.g., paraphrasing), leading to a high incidence of false negatives and thus limiting accuracy (Xu et al., 2025c). Further, this issue can become more extreme in long-form workloads, rendering such methods impractical. Fortunately, the emergence of generative reward models (Mahan et al., 2024; Zhang et al., 2025) alleviates the robustness problem. However, their current use in long-form workloads is typically based on pairwise or listwise preference ranking (Li et al., 2025b), which makes the reward unverifiable.

Reward verifiability is a shared objective across both search-augmentation workloads. Recent work on rubric-based rewards suggests creating verifiable generative signals by defining structured, interpretable criteria (Huang et al., 2025b; Gunjal et al., 2025). However, under long-form workloads, questions often seek information from multiple aspects, making the extraction of rubrics challenging. Moreover, due to the dynamic nature of real-world web corpora, these rubrics are difficult to maintain stable over time. Recent attempts avoid such issue by constructing rubrics along general dimensions (e.g., textual fluency, report completeness), but still remain vulnerable to reward hacking. This raises an important question: *what constitutes a verifiable rubric in the context of search augmentation?* Beyond the challenge of defining rubrics, practical deployment is further constrained because generative rewards require substantial computational resources and potentially introduce throughput bottlenecks in the RL pipeline (Li et al., 2025b; Wu et al., 2025), limiting their scalability in real-world applications.

In this paper, we propose a unified perspective for constructing verifiable generative rewards for both workloads. We consider the atomic golden information points (also known as *nuggets*[2]) as rubrics. This is aligned with the goal of search-augmentation, which is to correctly and comprehensively search for and output information that can solve the question, making the reward hack-resistant. Under this *nugget-as-rubric* paradigm, short-form workloads can be regarded as involving a single rubric, whereas long-form workloads expand the number of rubrics in proportion to the information demands of the question. By judging the entailment between the generated output and the defined rubrics, the rewards can be calculated in a verifiable way. Notably, the verifiable *nugget-as-rubric* paradigm is simple to implement for short-form workloads for many datasets provide explicit ground truth (Yang et al., 2018). For long-form workloads, we design an automatic rubric construction

---

[2]Following Pradeep et al. (2024b), a nugget refers to a complete unit-level claim or fact. In practice, a nugget is typically a 10–20 word declarative statement that includes a specific subject, an event description, and any relevant conditions or qualifiers.

pipeline that traverses the static corpus or online webs driven by query rewriting and exhaustively mines question-relevant passages until convergence, which replaces the typically incomplete and costly manual annotations methods (Arabzadeh et al., 2022). The gathered passages are then processed for nugget extraction, which comprises low-quality filtering, similarity-based merging, and weight assignment, confirming the usability of the *nugget-as-rubric* framework.

To efficiently verify rubrics, we train a 4B-parameter generative verifier, **Search-Gen-V**. Our method is based on the idea of distillation. First, a teacher verifier with large-scale parameters is selected to generate gold rubric verification labels in LLM-generated answers. Guided by the teacher verifier, we adopt a two-stage training procedure consisting of SFT and RL. We conduct multiple experiments to evaluate Search-Gen-V, including evaluation on validation set, short-form workload exemplified by HotpotQA (Yang et al., 2018), and long-form workload represented by DeepResearch Bench (Du et al., 2025). The results show that the Search-Gen-V-4B can significantly improve rubric verification across different settings, achieving performance comparable to the verifier model with over 200B parameters.

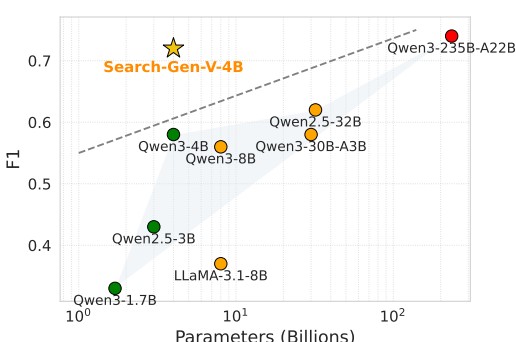

Figure 2: Our Search-Gen-V achieves a favorable balance between efficiency and performance in verifying rubrics for long-form answers.

To summarize, our main contributions include:
(i) we propose a unified perspective of verifiable generative reward paradigm for different workloads of search-augmented LLMs, which takes nuggets as rubrics; (ii) we design an automatic rubrics construction pipeline which replaces manual annotation, enabling a more comprehensive extraction of nuggets; (iii) we train a 4B efficient rubric verifier and demonstrate its effectiveness across short-form and long-form workloads.

## 2 RELATED WORK

**Search-augmented LLMs.** Search-augmentation refers to equipping LLMs with external retrieval capabilities, enabling them to access up-to-date and long-tail knowledge (Lewis et al., 2021; Gao et al., 2024). In this paradigm, generated outputs are no longer constrained by potentially hallucinatory internal knowledge, thereby improving factuality and trustworthiness (Jin et al., 2024; Wang et al., 2025). Current ways for search augmentation include traditional single-turn retrieval and multi-turn agentic retrieval (Jin et al., 2025; Xu et al., 2025b), and they are primarily applied to two types of tasks: short-form QA (Yang et al., 2018; Wei et al., 2024; 2025), where answers are usually single entities, and long-form QA (Du et al., 2025; Craswell et al., 2025a;b;c), where answers require the integration of multiple evidence points to produce paragraph-level or report-level outputs. Conventional methods typically rely on SFT to enhance search-augmented LLMs. Schick et al. (2023) employ SFT to train LLMs to invoke retrieval modules at appropriate stages. RA-DIT (Lin et al., 2024) and RankRAG (Yu et al., 2024) combine SFT with instruction tuning to improve LLMs' ability to exploit retrieved contexts. However, SFT-based methods face challenges in scaling with data size and risk trapping in a "memorization" pitfall (Chu et al., 2025).

**Reinforcement Learning for Search.** More recently, a line of work explores training search-augmented LLMs with RL (Jin et al., 2025; Song et al., 2025; Gao et al., 2025), which often yields better generalization. Widely adopted RL algorithms include PPO (Schulman et al., 2017), GRPO (Shao et al., 2024), and related variants. At the core of RL lies the design of reward signals. In RLVR, the reward signal is typically derived from verifiable rules or programmatic automatic checkers (Lambert et al., 2024; Yue et al., 2025), which can produce rewards that are objective, reproducible, and resistant to reward hacking. Such approaches are especially applicable in domains where correctness can be automatically verified, such as mathematical reasoning (Shao et al., 2024) and code generation (Dou et al., 2024). In the scenario of search, rule-based rewards such as Exact Match (Jin et al., 2025) or F1 score (Song et al., 2025) are verifiable but suffer from poor robust-

ness and cannot scale to long-form workloads. To address this, some studies employ generative reward models (Li et al., 2025b; Wu et al., 2025), which offer greater robustness. However, relying on generative models to perform pairwise preference judgments renders the reward non-verifiable and makes it vulnerable to hacking. Moreover, generative rewards are computationally expensive and may severely limit RL throughput. Thus, there is a lack of a reward paradigm that can provide signals that are simultaneously robust, verifiable, and efficient for search-augmented LLMs.

## 3 METHODOLOGY

### 3.1 NUGGET-AS-RUBRIC: DEFINITIONS FROM A UNIFIED PERSPECTIVE

First, we define the generation of search-augmented LLMs. Given a question $q$, the policy model $\pi_\theta$ (typically an LLM) invokes a search engine $\mathcal{R}$ in either a single-round or multi-round manner, and ultimately integrates the retrieved information to produce an predicted output $\hat{y}$. Formally,

$$\hat{y} \sim \pi_\theta \left( \cdot \mid q; \mathcal{R} \right). \tag{1}$$

While the definition is consistent, the form of $y$ differs between short-form and long-form workloads.

Now we introduce the concept of rubric-based reward. For each question $q$, there is an associated set of rubrics $\mathcal{R}$, representing multiple critic dimensions along which the prediction $\hat{y}$ to $q$ can be evaluated. Formally,

$$\Upsilon \left( q \right) = \{ \left( w_1, r_1 \right), \left( w_2, r_2 \right), \dots, \left( w_k, r_k \right) \}, \tag{2}$$

where $w_i \in \mathbb{R}$ indicates the weight of rubric $r_i$.

Although the form of $y$ varies for short-form and long-form workloads, we argue that both can be unified from the perspective of nugget (golden information unit). For short-form workload, the ground truth answer typically consists of a single entity, which can be regarded as a single nugget: $y^{\text{short}} \longrightarrow \{r_0\}$. In contrast, long-form workload requires answers covering multiple aspects, corresponding to multiple nuggets: $y^{\text{long}} \longrightarrow \{r_0, r_1, \dots\}$. Furthermore, since search-augmented LLMs aim to recall factoids faithfully, nuggets can fit this goal with unified form, verifiability, and hacking resistance, serving as the most appropriate instantiation of rubrics.

When constructing a verifiable reward based on *nugget-as-rubric*, we first need to verify whether each rubric is satisfied in the predicted output $\hat{y}$. We define a generative verifier model, $V_\varphi$, which takes as input a question $q$, a predicted output $\hat{y}$, and a rubric $r_i$, and produces a judgment $V_\varphi \left( q, \hat{y}, r_i \right) \in \mathbb{R}$, indicating whether $r_i$ is matched in $\hat{y}$. The judgment can be either continuous or discrete, such as a binary decision. Subsequently, we employ explicit rubric aggregation to compute the verifiable reward for the predicted answer, which can be calculated as:

$$R_\phi \left( q, \hat{y} \right) = \frac{\sum_{i=1}^{k} w_i \cdot V_\varphi \left( q, \hat{y}, r_i \right)}{\sum_{j=1}^{k} w_j}. \tag{3}$$

Then the reward can be used in RL to train the policy model $\pi_\theta$ for search-augmented generation, as illustrated below:

$$\max_{\pi_\theta} \mathbb{E}_{q \sim \mathcal{D}, \hat{y} \sim \pi_\theta(\cdot|q;\mathcal{R})} \left[ R_\phi \left( q, \hat{y} \right) \right] - \beta D_{\text{KL}} \left[ \pi_\theta \left( \hat{y} \mid q; \mathcal{R} \right) \| \pi_{\text{ref}} \left( \hat{y} \mid q; \mathcal{R} \right) \right], \tag{4}$$

where $\pi_{\text{ref}}$ is the reference model.

### 3.2 AUTOMATIC RUBRICS CONSTRUCTION

Rubrics construction is a prerequisite for implementing verifiable rubric-based rewards. For short-form workloads, rubrics construction is basically a simple challenge, since many manually annotated or synthetic datasets (Yang et al., 2018; Xu et al., 2025a) already provide a large amount of training data with short-form ground truth. In contrast, acquiring rubrics for long-form workloads remains tough. Nugget-based rubrics are often built on a set of passages associated with the question. Traditionally, organizations such as NIST rely on human annotators to identify relevant passages (Pradeep et al., 2024a). However, this approach is not only costly but, more critically, it depends on passage pooling, which labels only a small subset of top-ranked retrieval results. This introduces pool bias, which makes it highly likely to miss valid nuggets, ultimately leading to distorted rewards.

Figure 3: Illustration of the pipeline of our rubric-based verifiable reward modeling, which consists of two parts. Left (§3.2): automated generation of nugget-based rubrics. Right (§3.3): rubric verification using Search-Gen-V, which ultimately produces the reward.

Therefore, we propose an automated rubrics construction pipeline. Given a corpus $\mathcal{C}$, for a long-form question $q$, we define the oracle set of all passages relevant to $q$ as:

$$P(q) = \{p_1, p_2, \dots\}, \quad p_i \in \mathcal{C}. \tag{5}$$

We use MS MARCO V2.1 (Pradeep et al., 2024a), a large-scale corpus of real-world web pages. To better capture fine-grained information nuggets, we segment the corpus into passages, where each passage consist of 5-10 sentences. This segmentation strategy yields retrieval units of a manageable length, which are more suitable for handling by an LLM-based Judge, denoted as $\Psi$. It is worth noting that our pipeline can be readily adapted to other corpora, including dynamic web content. Finally, we adopt a dense retriever $E$, which indexes all passages in $\mathcal{C}$ for subsequent search operations.

---

**Algorithm 1:** Relevant Passages Mining in Automated Rubrics Construction

**Input:** Segmented corpus $\mathcal{C}$; question $q$; retriever $E$; LLM-based Judge $\Psi$

**Output:** Relevant passage set $P(q)$

Initialize passage set $\mathcal{P} \leftarrow \emptyset$, pending queue $\mathcal{W} \leftarrow \emptyset$;

Initialize search tree $T = (\mathcal{N} = \emptyset, \mathcal{E} = \emptyset, \text{root} = q)$;

$\mathcal{P}_0 \leftarrow \{p \mid p \in E(q; \mathcal{C}), \Psi(p; \text{Time}) = \text{True}\}$;

$T \leftarrow (\mathcal{N} \cup \{p\}, \mathcal{E} \cup \{(q, p) \mid\}, \text{root}), \quad p \in \mathcal{P}_0$;

$\mathcal{W} \leftarrow \mathcal{W} \cup \mathcal{P}_0$;

**while** $\underline{\mathcal{W} \neq \emptyset}$ **do**

  Pop $p_t$ from $\mathcal{W}$;

  $q_t \leftarrow \text{parent}(T, p_t)$;

  $\mathcal{Q}_i \leftarrow \{q' \mid q' \in \Psi(q_t, p_t; \text{Rewrite}), q' \notin \mathcal{N}\}$;

  $T \leftarrow (\mathcal{N} \cup \{q'\}, \mathcal{E} \cup \{(p_t, q')\}, \text{root}), \quad q' \in \mathcal{Q}_i$;

  **foreach** $\underline{q' \in \mathcal{Q}_i}$ **do**

    $\mathcal{P}_{q'} \leftarrow \{p \mid p \in E(q'), \Psi(p; \text{Time}) = \text{True}\}$;

    $\mathcal{P}_{q'} \leftarrow \{p \mid p \in \mathcal{P}_{q'} \mid p \notin \mathcal{N}\}, \mathcal{W} \leftarrow \mathcal{W} \cup \mathcal{P}_{q'}$;

    $T \leftarrow (\mathcal{N} \cup \{p\}, \mathcal{E} \cup \{(q', p)\}, \text{root}), p \in \mathcal{P}_{q'}$;

**return** $\underline{P(q) = \{p \mid p \in \mathcal{N}, p \text{ is passage}\}}$;

---

To mitigate pool bias, we adopt an iterative information mining approach based on query rewriting, aiming to exhaustively explore the boundary of $P(q)$. We leverage each retrieved passage as evidence to construct rewritten queries through entity substitution or constraint modification. Entity substitution involves synonym/hypernym/hyponym replacement, and constraint modification includes altering temporal, spatial, topical, or conditional constraints. This explicit, rule-guided rewriting method prevents $\Psi$ from generating ungrounded queries that might deviate from the actual requirements, and enables semantic-level expansion to cover potentially relevant information. Macroscopically, the entire process can be abstracted as the construction of a tree structure, where the nodes alternate between queries and passages. A query node has as its children the passages retrieved by that query, while a passage node has as its children the queries rewritten based on that passage. We define two types of stopping criterion for each path: (i) for a query node, the process terminates if no new, previously unseen passages can be retrieved; (ii) for a passage node, the process terminates if all rewritten queries are deemed similar to queries already present in the tree.

One critical issue to consider is the potential temporal misalignment between $q$ and the information contained in $\mathcal{C}$. Since static corpora often cover only a limited time span, "seemingly relevant" passages might be recalled. For example, considering the question "*What updates does the iPhone 17 Pro camera module have?*", if $\mathcal{C}$ contains only information prior to the release of iPhone 17, it may incorrectly retrieve passages that are semantically related but factually irrelevant. To address this, our pipeline performs a temporal consistency check for each passage–query pair. A passage is discarded if it fails to satisfy the explicit or implicit temporal constraints of the query, or if no causal relationship can be established between the passage and the query under temporal misalignment. In

addition, while retrieval typically selects the top-ranked passages for each query, the actual number of relevant passages may vary across different queries. Thus we leverage available query–passage relevance labels to conduct a statistical analysis of similarity scores obtained from $E$. We then determine a threshold score that distinguishes relevant from irrelevant passages, thereby improving the recall of relevant passages.

After estimating $P(q)$, we proceed to extract the nuggets of $q$ as rubrics $\Upsilon(q)$. We traverse each passage node in the tree and prompt $\Psi$ to extract nuggets. Each nugget is defined as a semantically complete factual statement (typically a sentence of about 10–20 words) that contributes to answering the question. Since the retrieval process aims to approximate the boundary of $P(q)$, it inevitably brings in noisy passages. To filter out low-quality candidates, $\Psi$ automatically verifies whether a nugget can establish a solid connection with the original question $q$. Moreover, because web corpora inherently contain similar or even duplicate content, nuggets extracted from different passages are further consolidated through similarity-based merging. Finally, we assign weights to the merged nuggets. Following the practices (Pradeep et al., 2024b; Xu et al., 2025b), we adopt a binary scheme: "vital", indicating that the nugget is highly important and must be included in the answer; and "okay", indicating that the nugget contains useful but non-essential information for the question.

### 3.3 SEARCH-GEN-V: AN EFFICIENT RUBRIC VERIFIER

We train a lightweight LLM as the rubric verifier, referred to as **Search-Gen-V**. To enable the verifier to scale across outputs of arbitrary length, we adopt an segmentation strategy. Specifically, for long-form workloads, the answer will be divided into blocks, where each block corresponds to a paragraph containing multiple claims and will be judged by all rubrics. To enhance efficiency, the verifier can examine multiple rubrics in a batch simultaneously. Each rubric is assigned a ternary label: (i) `support`, the rubric is fully satisfied in the block; (ii) `partially support`, the rubric is partially satisfied in the block; (iii) `not support`, the rubric is not satisfied at all. Finally, we apply a max-pooling strategy to aggregate rubric verification results across all blocks, and substitute the aggregated outcomes into Equation 3 to compute a verifiable reward.

We train Search-Gen-V through distillation from a teacher verifier. We compare two large-scale LLMs with different strategies: (i) Gemini-2.5-Flash (Gemini, 2025), which performs short reasoning and directly outputs the predicted label; (ii) Qwen3-235B-A22B-Instruct-2507 (Team, 2025), which adopts a voting-based method by picking the label with the most votes and, in the case of a tie, the more conservative option. A manual inspection shows that the first setting yields 24.9% of labels are more consistent with human judgments compared to the second setting, and we thus adopt it as the teacher verifier to produce teacher labels for supervising the training of Search-Gen-V.

We employ a two-stage training approach, consisting of SFT and RL. For robustness, we instruct the teacher verifier to output predicted labels in 10 different formats, such as Markdown, JSON. Further, we have the verifier also learn from the reasoning content generated by the teacher verifier in both stages. In the RL stage, we define a composite reward with the following components:

- *Prediction accuracy reward* (70%): This measures the agreement between the predicted label and the teacher label. We combine two complementary rewards, which are (i) Macro F1 score (35%) calculated between predicted and teacher labels, and (ii) Exact Match (35%), which equals 1 if the predicted labels exactly match the teacher labels and 0 otherwise.

- *Reasoning format reward* (20%): We allow the verifier to produce reasoning via an instruction-guided short format, enhancing efficiency over its intrinsic chain-of-thought mode. If the reasoning is generated in the form `<reasoning> ... </reasoning>` and contains substantive content, the reward is 1; if the format is correct but empty, the reward is 0.5; otherwise (incorrect format or missing final label), the reward is 0.

- *Output format reward* (10%): This checks whether the model outputs the predicted labels in the prescribed format. A correct format yields 1, otherwise 0.

We then train using the Decoupled Clip and Dynamic Sampling Policy Optimization (DAPO; Yu et al., 2025) algorithm, by maximizing the following objective function:

$$\mathcal{J}_{\text{DAPO}} = \mathbb{E}_{\left(q,\Upsilon,b,\{\ell_1,\dots,\ell_{|\Upsilon|}\}\right)\sim\mathcal{D}_{\text{train}},\{O_i\}_{i=1}^{G}\sim\pi_{\varphi_{\text{old}}}(\cdot|q,\Upsilon,b)}$$

$$\left[\frac{1}{\sum_{i=1}^{G}|O_i|}\sum_{i=1}^{G}\sum_{t=1}^{|O_i|}\min\left(\rho_{i,t}\left(\varphi\right)\hat{A}_{i,t},\text{clip}\left(\rho_{i,t}\left(\varphi\right),1-\varepsilon_{\text{low}},1+\varepsilon_{\text{high}}\right)\hat{A}_{i,t}\right)\right],$$

(6)

where $b$ denotes a block, $\ell_j$ denotes the gold label of rubric $r_j$, and $O_i$ contains the labels predicted by the verifier, i.e., $\left\{\ell'_{i,1},\dots,\ell'_{i,|\Upsilon|}\right\}\sim O_i$, and:

$$\rho_{i,t} = \frac{\pi_{\varphi}\left(O_{i,t}\mid q,\Upsilon,b,O_{i,<t}\right)}{\pi_{\varphi_{\text{old}}}\left(O_{i,t}\mid q,\Upsilon,b,O_{i,<t}\right)}, \quad \hat{A}_{i,t} = \frac{R_{\phi}^{i} - \text{mean}\left(\left\{R_{\phi}^{j}\right\}_{j=1}^{G}\right)}{\text{std}\left(\left\{R_{\phi}^{j}\right\}_{j=1}^{G}\right)}.$$

(7)

Following DAPO, we incorporate an overlength penalty, which introduces a soft penalty region, where responses that slightly exceed the ideal length receive a gradually increasing penalty, rather than an abrupt drop. The overlength penalty can be described as:

$$R_{\text{length}}(\hat{y}) = \begin{cases} 0, & |\hat{y}| \leq L_{\max} - L_{\text{cache}} \\ \frac{(L_{\max} - L_{\text{cache}}) - |\hat{y}|}{L_{\text{cache}}}, & L_{\max} - L_{\text{cache}} < |\hat{y}| \leq L_{\max} \\ -1, & L_{\max} < |\hat{y}| \end{cases}$$

(8)

Additionally, the dynamic sampling filters out judgments whose rubrics verification accuracy is 1 or 0, which is satisfying the following condition:

$$\text{s.t.} \quad 0 < \left|\left\{\ell'_{i,j} \mid \ell'_{i,j} \sim O_i, \ell'_{i,j} = \ell_{i,j}\right\}\right| < 1$$

(9)

## 4 EXPERIMENTS

In this section, we conduct a series of experiments to evaluate the performance of Search-Gen-V under different workloads, with the primary objective of verifying its ability to correctly generate rubric-based judgment labels for answers with respect to the corresponding questions.

### 4.1 EXPERIMENTAL SETUP

**Implementation Details.** In the rubric construction pipeline, we employ gte-modernbert-base (Zhang et al., 2024) as the retriever $E$, using Pyserini (Lin et al., 2021) for corpus indexing. Qwen3-235B-A22B-Instruct-2507-FP8 is used as the LLM-based judge $\Psi$. For the rubric verification stage, Qwen3-4B-Instruct-2507 serves as the base model for Search-Gen-V. Both the SFT and RL training stages are implemented using VeRL (Sheng et al., 2025). We then select two datasets of long-form workloads to construct the training data. First, the TREC Deep Learning Track dataset (Craswell et al., 2025a;b;c), which contains 207 questions with qrels, allowing us to directly apply nuggets extraction described in §3.2 to construct rubrics. Second, Researchy Questions (Rosset et al., 2024), from which we sample 3,000 questions and apply the full rubrics construction pipeline. Next, we employ six different search-augmented LLMs (from the Qwen and LLaMA series) to generate predicted answers for the above questions. Gold labels for rubric satisfaction in these long-form answers are then generated using the teacher verifier. Details provided in Appendix B and C.

**Workloads and Baselines.** We design our experiments from three settings: (i) *Validation set evaluation*. We utilize 84 available questions from the TREC RAG24 test split (Pradeep et al., 2024a), whose format is consistent with the training data. This is intended to evaluate the effectiveness of the training method of Search-Gen-V, and to serve as a bridge between long-form and short-form workloads. (ii) *Short-form workload*. HotpotQA and TriviaQA (Joshi et al., 2017) are chosen as representative short-form workloads. For each dataset, We sample 1,000 instances from its validation

Table 1: Results on the validation set. Rubric-level refers to the judgment accuracy of each rubric, while Sample-level refers to the accuracy of the aggregated labels across blocks. All metrics are macro-averaged over the ternary labels. We treat Qwen3-235B-A22B-Instruct-2507 as an oracle baseline, and the bold font highlights the best-performing verifier apart from the oracle baseline.

| Verifier Model | Rubric-level | | | Sample-level | | | Avg. F1 |
|---|---|---|---|---|---|---|---|
| | Precision | Recall | F1 | Precision | Recall | F1 | |
| Qwen3-1.7B | 0.41 | 0.49 | 0.34 | 0.48 | 0.40 | 0.32 | 0.33 |
| Qwen2.5-3B | 0.42 | 0.47 | 0.43 | 0.49 | 0.46 | 0.43 | 0.43 |
| Qwen3-4B | 0.56 | 0.62 | 0.57 | 0.61 | 0.58 | 0.58 | 0.58 |
| Qwen3-8B | 0.54 | 0.66 | 0.55 | 0.62 | 0.61 | 0.57 | 0.56 |
| LLaMA-3.1-8B | 0.45 | 0.54 | 0.42 | 0.34 | 0.41 | 0.32 | 0.37 |
| Qwen3-30B-A3B | 0.56 | 0.66 | 0.56 | 0.63 | 0.62 | 0.62 | 0.58 |
| Qwen2.5-32B-Instruct | 0.60 | 0.67 | 0.60 | 0.67 | 0.68 | 0.64 | 0.62 |
| Search-Gen-V-1.7B (SFT) | 0.63 | 0.62 | 0.62 | 0.66 | 0.66 | 0.66 | 0.64 |
| Search-Gen-V-4B (SFT) | 0.70 | 0.66 | 0.68 | 0.72 | 0.72 | 0.71 | 0.70 |
| Search-Gen-V-4B (SFT+RL) | **0.71** | **0.68** | **0.70** | **0.74** | **0.74** | **0.73** | **0.72** |
| Qwen3-235B-A22B-Instruct-2507 | 0.72 | 0.73 | 0.73 | 0.76 | 0.76 | 0.76 | 0.74 |

set and generate answers using different search-augmented LLMs. (iii) *Long-form workload*. This is an evaluation dataset focused on deep research. Its questions require in-depth exploration and integration of information from multiple sources, making them typical and challenging examples of long-form answer workloads. For detailed evaluation procedures, please refer to the following subsections and Appendix B.

## 4.2 VALIDATION RESULT

We construct rubrics for the questions in the TREC RAG24 test split using MS MARCO V2.1 corpus. Since qrels are available for these questions, only the nuggets extraction procedure in §3.2 for rubrics construction is required. Next, we generate predicted long-form answers for these questions using various search-augmented LLMs and split them into blocks. The teacher verifier is then used to produce ternary labels indicating the support of each rubric within these blocks, which serve as the gold labels. Upon analyzing these gold labels, we observe an imbalance issue. We thus apply data augmentation to address it, and details are provided in Appendix B.

We then compare the performance of Search-Gen-V with other baselines, as summarized in Table 1. We evaluate the verifier from two perspectives: rubric-level, which measures whether the support status of individual rubrics is correctly predicted, and sample-level, which assesses the correctness of the aggregated rubric support after combining block-level results. In Table 1, it can be observed that Search-Gen-V-4B outperforms all other baselines in both settings and achieves performance close to that of the large-scale verifier model, Qwen3-235B-A22B-Instruct-2507. Furthermore, we conduct an ablation study on the two-stage training, showing that both the SFT and RL stages contribute to performance gains. We also try using Qwen3-1.7B as the base model, however its SFT performance consistently fell short of expectations and was not comparable to the other baselines.

## 4.3 LONG-FORM WORKLOAD RESULT

Questions in the DeepResearch Bench require complex retrieval and reasoning to generate multi-faceted reports. We employ the same procedure as in the validation experiment in §4.2 to obtain rubrics. However, relevant information for these questions may not be present in the static corpus. Therefore, we integrate real-time Internet data into the rubrics construction pipeline, implemented via DuckDuckGo [3] and the Jina Reader API [4]. Then Search-Gen-V evaluates the support of each

---

[3] https://duckduckgo.com/
[4] https://jina.ai/reader/

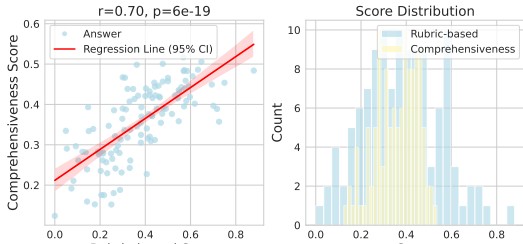

| Verifier Model | Precision | Recall | F1 |
|---|---|---|---|
| Qwen3-4B | 0.42 | 0.56 | 0.42 |
| Search-Gen-V-4B | **0.59** | 0.57 | 0.57 |
| Qwen3-235B-A22B | 0.57 | **0.67** | **0.61** |

Figure 4: Evaluation results of the long-form workload, DeepResearch Bench. Left figures: Rubric-based scores are generated by Search-Gen-V-4B. r denotes the Pearson correlation coefficient, and p indicates statistical significance. Right table: Accuracy comparison on verifying rubrics in long-form answers from DeepResearch Bench. All other settings are the same as in Table 1.

rubric with respect to answers, generating ternary labels. Nuggets labeled as "vital" are assigned a weight of 1, while "okay" nuggets receive a weight of 0.5. Each rubric judged as `support` contributes 1 point, `partially support` contributes 0.5 points, and `not support` contributes 0 points. Finally, a weighted average is computed using Equation 3 to produce the reward score.

To evaluate the utility of the score calculated by Search-Gen-V, we compare it with the Comprehensiveness metric (Du et al., 2025), as judged by Gemini-2.5-Pro. This metric assesses whether an answer covers key areas of the industry, ensures overall understanding, and avoids omitting important components, aligning with the objective of our proposed *nugget-as-rubrics* verifiable reward. We generate responses to 50 English questions in DeepResearch Bench using various deep research systems such as OpenAI DeepResearch (OpenAI, 2025) and, after filtering, obtain 119 valid long-form answers. We then compute the correlation between the two scores. As shown in Figure 4, the Pearson correlation coefficient reaches 0.7 and is statistically significant. And it achieves a substantial performance gains over the untrained 4B model and approaches the performance of Qwen3-235B-A22B. These results suggest that Search-Gen-V can serve as an open-source and efficient verifiable reward generator for more challenging long-form workloads.

In addition, to compare our method against more general preference-based reward modeling approaches, we adopt RewardBench 2 (Malik et al., 2025), a pairwise answer-preference benchmark commonly used to evaluate reward models.

Table 2: Results on RewardBench 2.

| Reward Model/Verifier Model | Type | Factuality Score |
|---|---|---|
| LMUnit-qwen2.5-72b | Generative | **87.2** |
| Skywork-Reward-V2-Llama-3.1-8B | Classifier | 84.6 |
| Search-Gen-V-4B | Generative | 85.8 |

To align with our focus on search-augmented LLMs, we consider only the factuality score and extract the corresponding rubrics using the same procedure as in our main setup. We select LMUnit-qwen2.5-72B (Saad-Falcon et al., 2025) and Skywork-Reward-V2-Llama-3.1-8B (Liu et al., 2025) as baselines. As shown in Table 2, despite less parameters, Search-Gen-V-4B achieves comparable performance, demonstrating its strong and stable generalization ability.

## 4.4 SHORT-FORM WORKLOAD RESULT

We select HotpotQA and TriviaQA as the representative of short-form workloads, where questions typically require multi-step reasoning. We first employ various search-augmented LLMs, such as Search-R1 (Jin et al., 2025), to generate answers for these questions. The teacher verifier then is used to assign gold labels for each pair of predicted answer and rubric (i.e., the ground-truth answer). Note that in this workload, the rubric contains only a single entity name, so under our ternary judgment scheme, the `partially support` category rarely occurs. We thus remove it and reduce the task to binary judgment. Correspondingly, in the prompts of Search-Gen-V, we also remove any instruction of `partially support` to adapt the model to binary prediction.

We compare against a typical rule-based reward for this workload, Exact Match (EM), which performs strict matching between the predicted answer and the rubric. Additionally, we include a comparison with generative judgments based on a large-scale LLM. The results in Figure **??** demonstrate that Search-Gen-V-4B achieves accuracy comparable to both EM and Qwen3-235B-A22B.

| Verifier Model | Precision | Recall | F1 |
|---|---|---|---|
| Qwen3-4B | 0.64 | 0.69 | 0.59 |
| Search-Gen-V-4B | 0.66 | 0.70 | 0.63 |
| Qwen3-235B-A22B | **0.70** | **0.76** | **0.69** |

Figure 5: Results of short-form workload, evaluating on HotpotQA. Left: cases of verifying rubrics satisfaction, where EM misjudges all cases due to bad robustness, while Search-Gen-V provides correct labels. Right: comparison of judgment accuracy on the 585 samples misjudged by EM.

Table 3: Results on the short-form workload, HotpotQA and TriviaQA. The first four baselines are single verifiers, and the last three are hybrid verifiers. Evaluations are performed on the full test set.

| Verifier Model | HotpotQA | | | TriviaQA | | |
|---|---|---|---|---|---|---|
| | Precision | Recall | F1 | Precision | Recall | F1 |
| EM | 0.84 | **0.80** | **0.82** | 0.82 | 0.78 | **0.80** |
| Qwen3-4B | 0.83 | 0.70 | 0.71 | 0.74 | 0.66 | 0.69 |
| Search-Gen-V-4B | 0.86 | 0.76 | 0.77 | **0.83** | 0.75 | 0.78 |
| Qwen3-235B-A22B | **0.87** | 0.78 | 0.80 | **0.83** | **0.79** | **0.80** |
| EM + Qwen3-4B | 0.94 | 0.92 | 0.93 | 0.87 | 0.82 | 0.84 |
| EM + Search-Gen-V-4B | 0.95 | 0.93 | 0.94 | **0.93** | 0.88 | 0.90 |
| EM + Qwen3-235B-A22B | **0.96** | **0.94** | **0.95** | **0.93** | **0.89** | **0.91** |

Although EM attains high accuracy, many rubrics in the HotpotQA are relatively unambiguous (e.g., yes/no). To further evaluate, we extract the samples misjudged by EM and, re-assess them as shown in Figure 5, finding that Search-Gen-V achieves over 60% accuracy, approaching the performance of Qwen3-235B-A22B. Therefore, Search-Gen-V can serve as a remedial method for rule-based functions in short-form workloads, enabling more accurate yet efficient reward construction.

## 5  CONCLUSION

In this paper, we analyze the limitations of current reward modeling for search-augmented LLMs. Rule-based rewards often suffer from robustness issues, while generative rewards face challenges in verifiability and computational cost. To address these issues, we propose a paradigm of *nugget-as-rubric* verifiable generative rewards, which unifies reward modeling for both short-form and long-form workloads. By leveraging the grounded nature of nuggets, our approach mitigates the lack of robustness and vulnerability to reward hacking. In addition, since long-form workloads typically involve diverse and multi-faceted rubrics, we introduce an automatic rubrics construction pipeline. This approach replaces the traditional manual annotation process, which is both labor-intensive and prone to pool bias. Finally, to improve reward computation efficiency for alleviating resource constraints and avoiding throughput bottlenecks in RL pipeline, we utilize a two-stage strategy to train a 4B verifier, Search-Gen-V. Results across different workloads show that Search-Gen-V-4B achieves higher reward computation accuracy on par with larger verifier models, establishing Search-Gen-V as a general, robust, and efficient verifiable reward constructor for search-augmented LLMs.

## LIMITATIONS

Although our automated rubrics construction pipeline eliminates the need for manual annotation, its iterative nature and reliance on LLM-based judge may lead to relatively slow convergence. Our experiments show that, on average, constructing rubrics for a single question from Researchy Questions dataset takes about one to two hours, suggesting that improving the efficiency of rubrics construction is an important direction for future work. Moreover, while this paper demonstrates the

effectiveness of Search-Gen-V-4B across workloads of search-augmented LLMs, we have not yet integrated it into an RL training pipeline. Prior evidence shows that increased reward accuracy tends to lead to improved RL performance, making this a natural avenue for extension, where we may assess RL convergence speed and throughput. Finally, for each workload, we experiment with only one representative dataset. Other datasets may differ in terms of domain, style, and other features, and thus future research should broaden evaluation and testing to a wider range of datasets.

## ETHICS STATEMENT

Our work aims to provide more general and accurate reward signals to enhance training effectiveness, ultimately enabling better-performing search-augmented LLMs that can support information dissemination for human society. Throughout the design of methods, the execution of experiments, and the collection of data, we have maintained a rigorous scientific attitude and strictly adhered to intellectual property and related agreements. We have also reported the potential limitations of this study. All datasets used are harmless and publicly accessible, and all research activities were conducted without any potential risks or harms.

## REPRODUCIBILITY STATEMENT

The algorithms and experimental results presented in this paper are readily reproducible. For the automated rubrics construction algorithm in §3.2, the tree structure is straightforward to implement: through iterative loops that repeatedly invoke the LLM-based judge for rewriting and judgment, the nodes of the tree can be progressively refined. All prompt templates are provided in the Appendix C. For the training method in §3.3, we build on the well-maintained open-source VeRL framework, which offers clear interface definitions that facilitate the implementation of our training logic. Furthermore, all experiments in this work are conducted on open-source datasets and open-source models, and both the LLM-based judge and web access APIs are obtained from widely used commercial platforms and are easily accessible.

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

## A  USE OF LARGE LANGUAGE MODELS

The subjects of this work are LLMs, which are also involved in rubric synthesis and the construction of gold labels. In the writing of this paper, all content was written entirely by the authors themselves, and LLMs were only used for polishing in terms of fluency, conciseness, and formatting, and were not involved in the substantive content. In all other aspects, including method design and code development, we declare that no LLM assistance was used.

## B  IMPLEMENTATION DETAILS

### B.1  TEACHER VERIFIERS

We experiment with two approaches to construct the teacher verifier:

- *Gemini-2.5-Flash*: guided by carefully designed prompt (identical to the one used by Search-Gen-V) to determine whether each rubric is satisfied in the answer.
- *Qwen3-235B-A22B-Instruct-2507*: prompted in a similar way, but augmented with a voting strategy. For each rubric, the label with the highest vote count was selected. In the case of ties, we adopted a conservative policy, with the priority order being: `not support` > `partially support` > `support`.

To assess the relative quality of these two strategies, we conduct manual expert annotation on samples where their predictions diverged. The comparative results are shown in Table 4. We found that, even with the voting strategy, the labels produced by Qwen3-235B-A22B are less reliable than those from Gemini-2.5-Flash. Consequently, we select the first approach, Gemini-2.5-Flash, as our teacher verifier for generating gold labels.

Table 4: Voting results across different teacher verifiers.

|        | Gemini | Qwen | Tie |
|--------|--------|------|-----|
| Votes  | 281    | 225  | 31  |

### B.2  VERIFICATION FORMATS

To enhance the robustness and generalization of Search-Gen-V, we employ multiple output formats and randomly, uniformly sample them during training. Specifically, the formats we adopted are presented in Table 5.

### B.3  ANSWERS GENERATION

Since our work focuses on verification, it is necessary to rely on model-generated data in order to conduct rubric-based validation. To this end, under different experimental settings, we employ a variety of models to generate answers for the questions in the corresponding datasets. Specifically, the models we used include:

- Training & Validation: Llama3.1-8B-Instruct, Qwen2.5-7B-Instruct, Qwen2.5-32B-Instruct, Qwen3-8B, Qwen3-30B-A3B, Qwen3-32B. The retriever used is GTE-Modernbert-Base.
- Short-form Workload: Search-R1-3B, LLaMA-3.1-8B-Instruct, Qwen2.5-3B-Instruct, Qwen2.5-32B-Instruct. The retriever used is E5-Base-V2 (Wang et al., 2022).
- Long-form Workload: Claude-3.5-Sonnet (with search), Claude-3.7-Sonnet (with search), Claude-Research, Doubao-DeepResearch, Gemini-2.5-Flash, Gemini-2.5-Pro, Gemini-2.5-Pro-DeepResearch, GPT-4.1, GPT-4.1-mini, GPT-4o, GPT-4o-mini, OpenAI-DeepResearch, Grok-DeepSearch, Kimi-Researcher, Langchain-Open-DeepResearch, Perplexity-Research, Sonar-Reasoning-Pro.

### B.4  TRAINING DATA AUGMENTATION

After using six different search-augmented LLMs to generate answers and constructing gold labels using the teacher verifier, the original distribution of the three labels is highly imbalanced: `support`

Table 5: Formats used in the training of Search-Gen-V.

| Format Name | Description | Example |
|---|---|---|
| JSON | Respond with a JSON array containing exactly one label for each nugget. | ["support", "not_support", "partial_support"] |
| csv | Respond with comma-separated values, one label for each nugget. | support,not_support,partial_support |
| Python List | Respond with a Python list containing exactly one label for each nugget. | ['support', 'not_support', 'partial_support'] |
| YAML | Respond with a YAML list, one label for each nugget. | support\n- not_support\n- partial_support |
| Markdown | Respond with a Markdown unordered list, one label for each nugget. | * support\n* not_support\n* partial_support |
| XML | Respond with XML format, one label for each nugget. | <labels>\n<label>support</label>\n<label>not_support</label>\n</labels> |
| tsv | Respond with tab-separated values, one label for each nugget. | support     not_support     partial_support |
| numbered | Respond with a numbered list, one label for each nugget. | 1. support\n2. not_support\n3. partial_support |
| comma-seperated | Respond with comma-separated values with spaces, one label for each nugget. | support, not_support, partial_support |
| pipe-seperated | Respond with pipe-separated values, one label for each nugget. | support\|not_support\|partial_support |

accounts for only about 9.76%, `partially support` about 5.49%, while `not support` dominates at 84.74%. Such severe imbalance can cause the model to be biased toward the majority class, reducing its ability to correctly identify minority classes and harming overall generalization. Thus, we conduct data augmentation to increase the proportion of `support` and `partially support` samples, enriching the diversity of training data and improving the model's ability to recognize minority classes and its robustness.

Each block corresponds to a set of rubrics categorized as `support`, `partially support`, and `not support`. During data augmentation, for each input to the model with a rubrics list length ranging from 1 to 10, all possible distributions of the number of nuggets per label are enumerated. For instance, if the list length is 3, the possible distributions include: (3,0,0), (2,1,0), (2,0,1), (1,2,0), (1,1,1), (1,0,2), (0,3,0), (0,2,1), (0,1,2), and (0,0,3), where the numbers represent counts of (`support`, `partially support`, `not support`). For each valid distribution, rubrics are randomly sampled from each label group and shuffled to form a new rubrics list. To control the dominance of `not support` nuggets, lists containing more than 5 rubrics with `not support` exceeding 50% are downsampled by randomly removing 20% of the `not support` rubrics while retaining all `support` and `partially support` rubrics. Due to the large number of augmented samples generated, a random 10% subset is selected as the final augmented dataset. After augmentation, the proportion of `support` rubrics increases from approximately 9.76% to 28.09%, `partially support` from 5.49% to 22.63%, and `not support` decreases from 84.74% to 49.28%, effectively increasing data diversity while mitigating label imbalance.

### B.5 DETAILS OF TRAINING

**SFT Training Hyperparameters.** During SFT, we use Qwen3-4B-Instruct-2507 as the backbone model. The training and validation sets are drawn from data-augmented and reasoning-enhanced datasets. The training batch size is set to 256, with a micro batch size per GPU of 2. The maximum input length is 8192 tokens. The learning rate is 1e-6 with a warm up ratio of 0.2. Weight decay is

0.1, and gradient clipping is 1.0. No LoRA is applied (`lora_rank=0`). We train for a total of 5 epochs, using 8 NVIDIA H100 GPUs per node.

**DAPO Training Hyperparameters.** For DAPO training, GRPO is adopted as the advantage estimator, and KL is disabled in reward computation. The policy LLM is trained with a learning rate of 1e-6, a generation batch size of 256, and a training batch size of 128. Maximum prompt and response lengths are 2048 and 4096 tokens, respectively, with left-side truncation. Filtering of generated batches is enabled, using up to 5 batches per group, optimized by the `seq_final_reward` metric. The actor model enables gradient checkpointing and bfloat16 precision. PPO mini-batch size is 64, with a maximum token length per GPU of 32k. KL loss is applied with a coefficient of 0.01 using `low_var_kl` type. Clip ratio is 0.28, gradient clipping is 1.0, entropy coefficient is 0.01, and loss is aggregated using token-level mean. Multi-turn rollout is enabled with a maximum of 1 assistant turn. Rollout temperature is 1.1, top-p is 1.0, top-k is disabled. For validation rollout, temperature is 0.7, top-p is 0.95, top-k is disabled, sampling is enabled, and one sample is generated per prompt. The reward model uses DAPO with an overlong buffer length of 2048 and a penalty factor of 1.0. Training is conducted on 8 NVIDIA H100 GPUs per node for a total of 800 steps.

## B.6 REWARD CURVES

We present the reward curves during the RL phase for both training and validation in Figure 6. We observe that under our combined reward design, the reward increases steadily without being dominated by the format reward. Moreover, the validation reward curve further demonstrates that the model is genuinely learning the intended verification behavior, rather than overfitting to formatting heuristics.

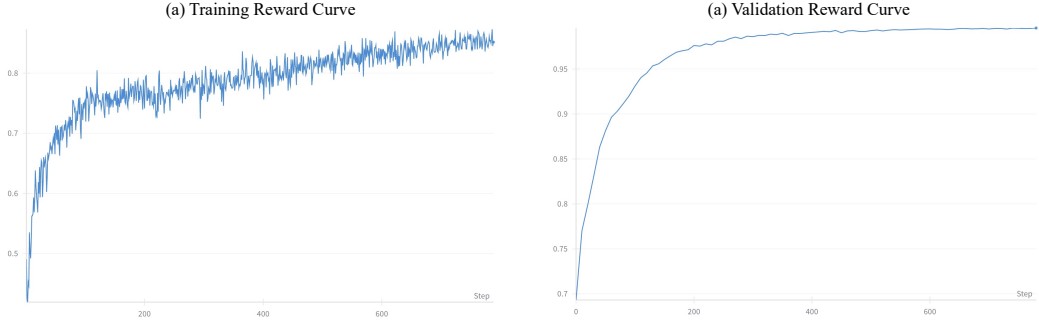

Figure 6: Reward curves during the RL phase of Search-Gen-V-4B.

## C PROMPT TEMPLATES

We provide all prompt templates used in the methods implementation and experiments of this work.

**Prompt templates of automatic rubrics construction pipeline.** Query rewriting based on a passages:

> You are an expert in query rewriting, able to write useful new queries based on relevant information.
>
> Task Description:
> Given a question, a query, and a passage, you need to generate new queries by modifying the given query based on the information in the given passage.
>
> Background:
> This is not a general query rewriting task; rather, it is a step in the task of mining ground truth information for the given question within a web corpus. The given question usually comes from

long-form QA datasets or research-style question datasets, which require multiple information points to answer. The given query was generated during the mining process, and the given passage is exactly what was retrieved using this query.

Core Principles:
1. The information referenced from the given passage is usually related entities and modifiers associated with the given query, which were not considered in the query itself.
2. The rewriting actions can only be selected from the given Executable Rewriting Operations, with a maximum of three operations combined per rewrite.
3. The rewritten query needs to be semantically expanded, making it more likely to recall passages that contain ground truth information for the question but have not yet been mined.
4. The rewritten query must remain strictly within the domain relevant to the given question, and must not introduce any unrelated queries.

Executable Rewriting Operations:
1. Synonym replacement
2. Hypernym replacement
3. Hyponym replacement
4. Entity name fuzzification
5. Entity name specification
6. Switching between interrogative forms such as what/how/why
7. Add or modify constraints on the query (i.e., time, location, topic, condition, etc.)

Output Format (two parts):
1) Short reasoning: Place ALL your reasoning analysis inside <reasoning> ... </reasoning> tags. You can freely express your thought process, but follow the steps below:
- Recall the information from the given passage that is useful for the rewrite.
- If there is useful/relevant information in the passage, analyze which rewriting operations need to be applied.
- Execute the rewrite.
PS: Do not generate <reasoning> or </reasoning> inside the <reasoning> ... </reasoning> tags to avoid parsing errors.

2) Generate the final rewritten queries: After the </reasoning> tag, provide the final rewritten queries. You need to follow the requirements below:
- Output one plain-text new query per line, with no other content.
- Generate at most max_num_new_queries rewritten queries.
- If no rewritten queries can be generated, output [None] directly.
- It is better to provide fewer or even zero queries than to include irrelevant or low-quality ones.

Question: {question}
Query to be rewritten: {query}
Passage: {passage}

Duplication checking whether the newly generated rewritten query is identical or similar to any existing queries:

You are an expert in search query judgment, capable of identifying similar queries.

Task Description:
Given a rewritten query and a batch of existing queries, you need to determine whether the rewritten query is similar to any of the existing queries.

Background:
This task is part of an information mining process through query rewriting. The goal is to determine whether a newly rewritten query is similar to an existing query, in order to avoid redundant retrieval. The strategies for query rewriting include synonym replacement,

hypernym/hyponym replacement, entity name fuzzification or specification, interrogative form transformation, modification or addition of constraints, and so on.

Core Principles:
1. The definition of "similar" is that the rewritten query shares the same entity names and constraints as an existing query.
2. Do not judge by deep semantics. Consider queries similar only if they look similar on the surface. For instance, "older people" and "elderly individuals" should be treated as different. Keeping such similar queries helps expand the semantic representation range of the retriever and thus avoid missing information.
2. If a query differs superficially from an existing query in terms of entity names or constraints but is semantically equivalent, it should also be considered similar. Such as "older people" and "elderly individuals".

Output Format (two parts):
1) Short reasoning: Place ALL your reasoning analysis inside <reasoning> ... </reasoning> tags. You can freely express your thought process to compare the newly rewritten query with each existing query whether they are similar. Do not generate <reasoning> or </reasoning> inside the <reasoning> ... </reasoning> tags to avoid parsing errors.

2) Generate the final decision: After the </reasoning> tag, provide the final decision. You need to follow the requirements below:
- If the rewritten query is similar to any query in the existing queries, return True;Otherwise, return False.
- Do not generate any other content.

The rewritten query: {rewritten_query}
A batch of existing queries: {existing_queries}

Verify whether the retrieved passage satisfies the temporal consistency requirements of the query:

You are a professional LLM Judge.

Task Description:
Given a query and a passage retrieved based on that query, you are asked to determine whether the passage satisfies the time constraint specified in the query.

Background:
This task is part of an information mining process through query rewriting. Since the topics being explored may differ from the creation time of the corpus, there is a risk of retrieving information that is temporally inconsistent with the query. The purpose of this task is to prevent the exposure of such information.

Output format (two parts):
1) Short reasoning: Place ALL your reasoning analysis inside <reasoning> ... </reasoning> tags. You can freely express your thought process, but follow the steps below:
- Check whether the query contains any temporal features. If no temporal features are present, or if the query accepts information across a broad time range, then any passage can be considered to satisfy the time constraint. End reasoning.
- If the query contains temporal features, determine the time scope of the query. Options include:
- A specific point in time (e.g. a particular year or century).
- A time range (which can be between two points in time, before a certain time, or after a certain time).
- Then, based on the intent of the query, determine the type of time constraint that the passage needs to satisfy. Options include:
- Strictly Constrained: The passage information must be strictly within the time range specified

by the query. For example, the query "floods in Asia in 2015" requires the passage to contain information strictly from 2015.
- Forward Time Extension: The passage may include information earlier than the time range specified by the query, emphasizing causes or background related to the query. For example, the query "what were the political causes of the 2015 oil crisis" accepts information from before 2015.
- Backward Time Extension: The passage may include information later than the time range specified by the query, emphasizing effects or consequences of the query. For example, the query "impact of the 2008 financial crisis on the automotive industry" accepts information from after 2008.
- Based on the determined type of time constraint, analyze whether the passage satisfies the corresponding requirement. End reasoning.
PS: Do not generate <reasoning> or </reasoning> inside the <reasoning> ... </reasoning> tags to avoid parsing errors.

2) Generate the final decision: After the </reasoning> tag, provide the final decision. You need to follow the requirements below:
- If the passage meets the time constraint specified in the query, output True; Otherwise, output False.
- Do not generate any other content.

Query: {query}
Passage: {passage}

Extract rubrics (nuggets) from a relevant passage:

You are NuggetCreator, an intelligent assistant that can generate atomic nuggets of information from a passage.

Task:
Given a question and a possibly relevant or useful passage, you need to generate atomic nuggets of information from the passage, so that the nuggets can be the gold information required to answer the question.

Core Principles:
1. Each generated nugget should be a complete and unique statement of a fact from the passage (a sentence of about 10-20 words).
2. A nugget should include a clear subject, verb, object, and if necessary, include constraint information such as time, location, topic, etc.
3. A nugget should avoid using pronouns such as "it".
4. A nugget is not simply a salient statement within the context, but also one that helps answer the question.

Output Format (two parts):
1) Short reasoning: Place ALL your reasoning analysis inside <reasoning> ... </reasoning> tags. You can freely express your thought process, but follow the steps below:
- Identify key factual statements in the passage.
- If there are complete statements, determine whether each factual statement is valuable in answering the given question by organizing the answer from multiple perspectives, and based on that, decide whether to consider it as a nugget.
PS: Do not generate <reasoning> or </reasoning> inside the <reasoning> ... </reasoning> tags to avoid parsing errors.

2) Generate the nuggets: After the </reasoning> tag, provide the nuggets. You need to follow the requirements below:
- Output one plain-text nugget per line, with no other content.
- Make sure you generate at most creator_max_nuggets nuggets (can be less or empty).

- If no complete statement that is valuable to the question can be found in the passage, do not generate any low-quality nuggets, and just return [None] directly.
- Do not explain and make sure there is no redundant information.

Question to be answered: {question}
Passage: {passage}

Merge duplicated or similar rubrics (nuggets):

You are NuggetMerger, an intelligent assistant that can combine similar nuggets.

Task:
Given a question and a list of nuggets (each nugget corresponds to a ID number), you need to combine similar nuggets if necessary.

Background:
A nugget refers to a semantically complete factual statement (a sentence of about 10-20 words) that helps answer the given question. A nugget should include a clear subject, verb, object, and if necessary, include constraint information such as time, location, topic, etc. Since there may be multiple sources containing similar information, the nuggets may be similar or even duplicated.

Core Principles:
1. "Similar" means that two or more nuggets point to the same factual statement at the semantic level.
2. Merge similar nuggets into a single nugget, making sure it is the best and most complete description of the factual statement.
3. When merging, ensure that the merged nugget is not too long (more than 20 words) and does not lose any useful information.

Output Format (two parts):
1) Short reasoning: Place ALL your reasoning analysis inside <reasoning> ... </reasoning> tags. You can freely express your thought process, but follow the steps below:
- Identify whether there are similar nuggets.
- If there are similar nuggets and merging them would not make the merged nugget too long (more than 20 words), group the nuggets that need to be merged together, and record the ID numbers of the nuggets in each group.
- For each group, merge and rewrite the nuggets into a single nugget.
PS: Do not generate <reasoning> or </reasoning> inside the <reasoning> ... </reasoning> tags to avoid parsing errors.

2) Generate the final merged nuggets: After the </reasoning> tag, provide the final merged nuggets. You need to follow the requirements below:
- Output one plain-text merged nugget per line, following the indication of the ID numbers of the original nuggets that are merged into it. Example: nugget_text [1, 2, ...]
- When nuggets are merged, the nuggets that are not merged should still follow the format of indicating their original ID numbers.
- If there are no similar nuggets in the list, which means that no merging is needed, simply return: [NO NEED].
- Do not explain and make sure there is no redundant information.

Question: {question}
List of nuggets:{nuggets}

Assign a weight to each rubric (nugget), either "vital" or "okay":

You are NuggetScorer, an intelligent assistant that can label a list of nuggets based on their importance to a question.

Task:
Given a question and a list of nuggets, you need to label each of the [{num_nuggets}] nuggets either a "vital" or "okay" based on the following core principles.

Background:
A nugget refers to a semantically complete factual statement (a sentence of about 10 words) that can be the gold information required to answer the given question.

Core Principles:
1. A "vital" nugget represents a factual statement that must be present in a "good" answer, whether it pertains to the overall question or a specific aspect.
2. An "okay" nugget contributes worthwhile information about the question but is not essential; in other words, it is "good to have" but not mandatory.

Output Format (two parts):
1) Short reasoning: Place ALL your reasoning analysis inside <reasoning> ... </reasoning> tags. You can freely express your thought process about the reasons why each nugget is "vital" or "okay". Do not generate <reasoning> or </reasoning> inside the <reasoning> ... </reasoning> tags to avoid parsing errors.

2) Generate the final labels: After the </reasoning> tag, provide the final labels. You need to follow the requirements below:
- Output the label of each nugget on a separate line.
- The label must be either vital or okay, in plain text only, with no other content.
- Do not explain and make sure there is no redundant information.

Question: {question}
List of nuggets: {nuggets}

**Prompt templates of rubric verification used by Search-Gen-V.** Verify the support status of a batch of rubrics in an answer-block, allowing reasoning, and output a ternary label:

You are NuggetMatchJudge.

Task:
Given a search query, a passage, and {num_nuggets} nuggets, assign one label to each nugget: "support","partial_support" or "not_support".

Core Principle:
Your judgment must be based EXCLUSIVELY on the provided passage. Do not use any external knowledge.

Label Definitions & Decision Process:
Please follow this decision framework for each nugget:
1. Check for Contradiction → "not_support"

  - Does the passage explicitly state the opposite of the nugget?

  - If yes, label "not_support".

2. Check for Full Support → "support"

  - Are ALL essential facts of the nugget explicitly and unambiguously stated in the passage?

  - Essential facts include: subjects, actions, key quantities, dates, conditions, and qualifiers

  - Do all qualifiers (e.g., "always", "some", "may") match perfectly?

- If yes, label "support".

3. Check for Partial Support → "partial_support"

   - Does the passage support at least one essential fact, but another essential fact is missing, hedged (e.g., "may", "suggests"), or stated ambiguously?
   - Does verifying the nugget require only a minor, safe inference (e.g., treating clear paraphrases like "reached the summit" as equivalent to "climbed the mountain")?
   - If yes, label "partial_support".
   - Safe inference example: Passage says "turnover of \$10 million", nugget says "revenue of \$10 million"
   - Unsafe inference example: Passage says "CEO bought a new car", nugget says "company is doing well financially"

4. Default → "not_support"

   - If none of the above conditions are met (information entirely absent or only topically related), label "not_support".

Output Format (two parts):
1) Reasoning: Place ALL your reasoning analysis inside <reasoning>... </reasoning>tags. For each nugget, freely express your thought process, including:

   - Restate the nugget to ensure understanding
   - Quote or paraphrase relevant parts from the passage
   - Analyze the relationship and support level
   - Reach a conclusion (support/partial_support/not_support)
     Use any format that helps you think clearly - paragraphs, bullet points, or numbered lists.

2) Final Answer: After the </reasoning>tag, provide the final labels in the requested format.

   - {format_instruction}
   - No extra text after the labels.
   - Before submitting the Final Answer, confirm 3 points:
     (1) Order matches nugget serial numbers;
     (2) No repeated labels for any nugget;
     (3) Number of labels = {num_nuggets}.
     Only submit if all 3 points are satisfied.

Search Query: {query}
Passage: {passage}
Nuggets ({number_nuggets}): {nugget_list}
Please provide your detailed reasoning in <reasoning>... </reasoning>tags, then collect the final result for each nugget from the reasoning section and list them in order:

**Prompt templates of rubric verification used by Search-Gen-V.** Verify the support status of a batch of rubrics in an answer-block, allowing reasoning, and output a binary label without `partially support`:

You are NuggetMatchJudge.

Task:
Given a search query, a passage, and {num_nuggets} nuggets, assign one label to each nugget: "support" or "not_support".

Core Principle:

Your judgment must be based EXCLUSIVELY on the provided passage. Do not use any external knowledge.

Label Definitions & Decision Process:
Please follow this decision framework for each nugget:
1. Check for Contradiction → "not_support"

   - Does the passage explicitly state the opposite of the nugget?

   - If yes, label "not_support".

2. Check for Full Support → "support"

   - Are ALL essential facts of the nugget explicitly and unambiguously stated in the passage?

   - Essential facts include: subjects, actions, key quantities, dates, conditions, and qualifiers

   - Do all qualifiers (e.g., "always", "some", "may") match perfectly?

   - If yes, label "support".

3. Default → "not_support"

   - If none of the above conditions are met (information entirely absent or only topically related), label "not_support".

Output Format (two parts):
1) Reasoning: Place ALL your reasoning analysis inside <reasoning>... </reasoning>tags. For each nugget, freely express your thought process, including:

   - Restate the nugget to ensure understanding

   - Quote or paraphrase relevant parts from the passage

   - Analyze the relationship and support level

   - Reach a conclusion (support/not_support)
     Use any format that helps you think clearly - paragraphs, bullet points, or numbered lists.

2) Final Answer: After the </reasoning>tag, provide the final labels in the requested format.

   - {format_instruction}

   - No extra text after the labels.

   - Before submitting the Final Answer, confirm 3 points:

     (1) Order matches nugget serial numbers;
     (2) No repeated labels for any nugget;
     (3) Number of labels = {num_nuggets}.

     Only submit if all 3 points are satisfied.

Search Query: {query}
Passage: {passage}
Nuggets ({number_nuggets}): {nugget_list}
Please provide your detailed reasoning in <reasoning>... </reasoning>tags, then collect the final result for each nugget from the reasoning section and list them in order:

**Prompt templates of HotpotQA answer.** A structured Q&A template that guides the model to reason first, optionally retrieve information, and then output a concise final answer:

You are a helpful and harmless assistant.

Answer the given question. You must conduct reasoning inside `<think>` and `</think>` first every time you get new information. After reasoning, if you find you lack some knowledge, you can call a search engine by `<tool_call>` query `</tool_call>` and it will return the top

> searched results between `<tool_response>` and `</tool_response>`. You can search as many times as your want. If you find no further external knowledge needed, you can directly provide the answer inside `<answer>` and `</answer>`, without detailed illustrations. For example, `<answer> Beijing </answer>`. Question:

# D EXAMPLES

## D.1 EXAMPLES OF RUBRICS

We select a question from the TREC RAG24 test set, "Why are people boycotting Starbucks?", and illustrate the rubrics constructed by our automatic rubrics construction pipeline, as shown below:

> Rubric 1: Starbucks CEO Howard Schultz expressed intolerance for traditional marriage supporters, leading to a boycott by anti-gay marriage groups. [vital]
>
> Rubric 2: Starbucks has been criticized for tax avoidance and failing the Fair Trade test. [vital]
>
> Rubric 3: Starbucks is under boycott due to its promotion of GMO agriculture and use of non-organic products. [vital]
>
> Rubric 4: Starbucks is being boycotted for its promise to hire 10,000 refugees. [vital]
>
> Rubric 5: Starbucks donated money to Planned Parenthood. [vital]
>
> Rubric 6: Starbucks supported a referendum backing gay marriage in Washington state. [vital]
>
> Rubric 7: Starbucks donated hundreds of thousands of dollars to Democrats. [vital]
>
> Rubric 8: Starbucks CEO took a stand against President Trump's executive order. [vital]
>
> Rubric 9: Conservative Christians called for a boycott of Starbucks last winter. [vital]
>
> Rubric 10: Some people are boycotting Starbucks because of the cups. [vital]
>
> Rubric 11: People boycotted Starbucks after two Black men were arrested. [vital]
>
> Rubric 12: Starbucks CEO Howard Schultz told a shareholder to sell his shares if he supported traditional marriage. [vital]
>
> Rubric 13: The National Organization for Marriage called for a boycott of Starbucks. [okay]
>
> Rubric 14: The boycott by traditional marriage supporters caused a drop in Starbucks sales revenue. [okay]
>
> Rubric 15: Individuals can boycott brands due to tax shaming. [okay]
>
> Rubric 16: Dice led a boycott of Starbucks due to its logo. [okay]
>
> Rubric 17: Starbucks closed stores nationwide for sensitivity training. [okay]
>
> Rubric 18: Donald Trump encouraged boycotting Starbucks while campaigning. [okay]
>
> Rubric 19: Starbucks has been criticized for its treatment of workers. [okay]
>
> Rubric 20: Conservatives urged a boycott of Starbucks over its minimalist red holiday cups. [okay]

