# OpenReview forum: "An Efficient Rubric-based Generative Verifier for Search-augmented LLMs"
_ICLR.cc/2026/Conference — Submitted to ICLR 2026_

### Official Review · Reviewer_r9Uu · 2025-10-27

**Soundness:** 3
**Presentation:** 3
**Contribution:** 3
**Rating:** 6
**Confidence:** 3

**Summary:**

This paper proposes a unified and verifiable paradigm, namely ``nugget-as-rubric", which treats atomic information points as structured evaluation criteria for different search-augmentation workloads. Short-form tasks correspond to a single rubric, whereas long-form tasks expand to multiple rubrics aligned with the question’s information needs. To support long-form settings, this paper designs an automatic rubric construction pipeline based on query rewriting, which can automatically retrieve passages relevant to each question and extract rubrics from them, both from static corpora and from dynamic online web content. Experimental results show that the proposed method and the trained model achieve strong verification accuracy across different workloads, making it a scalable, robust, and efficient verifiable reward constructor for search-augmented LLMs.

**Strengths:**

1. The paper proposes "nugget-as-rubric," a unified paradigm that treats atomic information points (nuggets) as structured evaluation criteria (rubrics). This approach successfully unifies the reward modeling for both short-form tasks (seen as a single rubric) and long-form tasks (seen as multiple rubrics). The method is designed to overcome the flaws of current reward models. It solves the "fragility" of rule-based rewards (like Exact Match), which perform poorly with variations in expression and cannot scale to long-form tasks. It also addresses the issues of generative rewards, which are often non-verifiable, unstable, and computationally expensive for long-form workloads.

2. The paper introduces an automatic rubric construction pipeline. This pipeline uses query rewriting to retrieve relevant passages and extract nuggets from both static corpora and dynamic web content. This automated process replaces traditional manual annotation, which is costly, labor-intensive, and prone to bias.

3. Experiments show that Search-Gen-V-4B achieves strong verification accuracy across different workloads. Notably, its performance is comparable to a much larger 200B+ parameter verifier model (Qwen3-235B-A22B-Instruct-2507) , making it a scalable, robust, and efficient verifiable reward constructor.

**Weaknesses:**

1. While the automated rubric construction pipeline eliminates manual annotation, its iterative nature and reliance on an LLM-based judge result in slow convergence. The authors state that constructing rubrics for a single question takes, on average, one to two hours.

2. The experiments for each workload (short-form and long-form) were conducted on only one representative dataset. The authors acknowledge that other datasets may have different characteristics, and future research should expand the evaluation to a wider range of datasets.

**Questions:**

None

---

> ### Author Response · Authors · 2025-11-26
> **Response to Reviewer r9Uu**
>
> Dear Reviewer r9Uu,
>
> Thank you for recognizing our work and for your valuable feedback. In this response, we address each of the weaknesses you have raised in detail.
>
> ## Weakness 1
>
> Indeed, the total time required to mine rubrics for a single query—on average 1–2 hours—may seem slow, as it involves performing BFS over a web-scale corpus of segments until convergence—i.e., until no new segments are judged to be useful. Although this may seem time-consuming, there are several mitigating factors:
>
> 1. The time depends on the complexity of the query; simpler queries can be completed in just tens of minutes.
> 2. Our implementation uses concurrent mining, which relies on the resources of the corpus retrieval server and the LLM-as-Judge, so the actual wall-clock time is not equivalent to serial runtime.
>
> In our experiments, we employed 100 workers, and under continuous operation for 24 hours, approximately 1.9k rubrics can be mined. This allows us to quickly accumulate a sufficient volume for training purposes.
>
>
> ## Weakness 2
>
> Thank you for your suggestion. In future work, we plan to extend our experiments to more datasets and scenarios for search-augmented LLMs. At present, however, in the long-form setting, the availability of suitable evaluation datasets is indeed limited. Well-known benchmarks such as GAIA and xBench-Deep Search adopt entity-based short-form answers, which are not aligned with the long-form verification task studied in our work. Currently, both DeepResearchBench and our own validation set produce long-form answers grounded in web retrieval, making them naturally suitable for evaluating rubric-based long-form verification. Therefore, expanding experiments under the long-form workload is challenging due to the scarcity of appropriate benchmarks. We kindly ask for your understanding regarding this limitation.
>
> As for short-form workloads, we have conducted an additional evaluation on TriviaQA. We sampled 1,000 examples from its validation set and generated answers using the same search-augmented LLMs and search tools as in our main experiments. The results are presented in Table 1. The findings show that Search-Gen-V-4B also achieves performance close to Qwen3-235B on this dataset, further demonstrating the generalizability of our reward model across different short-form benchmarks.
>
> | Verifier Model  | Precision | Recall | F1   |
> |-----------------|:---------:|:------:|------|
> | Qwen3-4B        |    0.74   |  0.66  | 0.69 |
> | Search-Gen-V-4B |    0.83   |  0.75  | 0.78 |
> | Qwen3-235B-A22B |    0.83   |  0.79  | 0.80 |
>
> ---
>
> Once again, we sincerely thank you for the time and effort you have devoted to reviewing our submission. We hope that our responses have addressed some of your concerns and look forward to further discussions.

---

### Official Review · Reviewer_oyh1 · 2025-10-30

**Soundness:** 3
**Presentation:** 2
**Contribution:** 3
**Rating:** 6
**Confidence:** 3

**Summary:**

This paper introduces Search-Gen-V, a rubric-based generative verifier for search-augmented LLMs. The key idea is to represent factual “nuggets” as structured rubrics that provide verifiable supervision for both short-form and long-form search tasks. Through an automated rubric-generation pipeline and a two-stage SFT + RL distillation process, a compact 4B-parameter verifier achieves performance comparable to much larger models on TREC, DeepResearchBench, and HotpotQA.

**Strengths:**

- Clear motivation and practical relevance: Addresses a genuine bottleneck in search-augmented LLMs—how to construct verifiable yet robust rewards for reinforcement learning with retrieval-based systems.

- The nugget-as-rubric formulation elegantly bridges short-form and long-form search workloads under a single paradigm, improving consistency across RL reward modeling.

- Search-Gen-V-4B is efficient as it achieves near-parity with 200B-scale models at significantly lower computational cost.

**Weaknesses:**

- Several core components (e.g., rubric aggregation, DAPO optimization schedule, interaction between SFT and RL stages) are insufficiently detailed for replication.

- The contribution mainly integrates existing ideas—rubric-based verification, nugget extraction, and reward distillation—into one pipeline rather than introducing a fundamentally new principle. The advantage of “nugget-as-rubric” over prior rubric or preference-based reward models (e.g., standard LLM judges) is not sharply articulated.

- The verifier is not yet used in an RL loop to show downstream improvements. It would be better to provide some end-to-end demonstration of reward effectiveness.

- No systematic study of the quality for rubrics.

**Questions:**

See above. Some additional questions:

- What is the runtime and cost of rubric generation per instance, and can it scale efficiently to large corpora?

- How do you detect and filter erroneous or hallucinated rubrics during automatic construction?

- How does the rubric weighting scheme influence performance? Have any learned aggregations been attempted?

---

> ### Author Response · Authors · 2025-11-26
> **Response to Reviewer oyh1 (part #1)**
>
> Dear Reviewer oyh1,
>
> Thank you for your positive assessment of our work and for your valuable feedback. In the following response, we address each of the weaknesses you raised in detail.
>
> ## Weakness 1
>
> We confirm that we will release our code, model checkpoints, and all training and evaluation data to the public community. For the review process, we have provided an anonymous link to the code. We kindly invite you to take a look.
>
> [link](https://anonymous.4open.science/r/ICLR-Rebuttal-C82D/README.md).
>
>
> ## Weakness 2
>
> While our work is indeed built upon prior foundational studies, our primary goal is to address several limitations that remain unsolved in existing approaches:
>
> - Ungrounded rubric construction. Previous methods for constructing rubrics can be ungrounded, which introduces a risk of reward hacking for search-augmented LLMs. To mitigate this issue, we treat nuggets as rubrics and develop an automated rubric construction pipeline to replace manual intervention.
> - Lack of generalization beyond short-form tasks. Prior search-augmented LLMs are predominantly optimized for short-form tasks. In contrast, our aim is to build a unified reward modeling framework that can generalize across domains and data sources, including transitions from short-form to long-form tasks.
> - High cost of rubric-based evaluators. Existing rubric-based evaluators often rely on large models, which impose substantial deployment costs. When such evaluators are used within actual RL training—especially dense reward settings with process-level feedback—they must also meet the throughput requirements of RL pipelines. This makes it crucial for the reward model to be as small as possible (4B) and highly efficient (supporting batched verification).
>
> In addition, we have additionally included comparisons with preference-based reward models. Specifically, we selected two reward models from RewardBench 2 [1]:
>
> - ContextualAI/LMUnit-qwen2.5-72b [2]
> - Skywork/Skywork-Reward-V2-Llama-3.1-8B [3]
>
> Our Search-Gen-V-4B continues to determine the chosen output using our rubric-based scoring method. We compare Search-Gen-V-4B against these two baselines on the factuality metric in RewardBench 2, which aligns more closely with the objectives of search-augmented LLMs. The results in Table 1 further demonstrate the necessity of training a dedicated reward model, especially a smaller and more efficient one, to support dense-reward RL training.
>
> Table 1:
> | Reward Model/Verifier Model            |    Type    | Factuality Score |
> |----------------------------------------|:----------:|:----------------:|
> | ContextualAI/LMUnit-qwen2.5-72b        | Generative |       87.2       |
> | Skywork/Skywork-Reward-V2-Llama-3.1-8B | Classifier |       84.6       |
> | Ours/Search-Gen-V-4B                   | Generative |       85.8       |
>
>
> ## Weakness 3
>
> Thank you for your insightful suggestion. In this work, we focus specifically on the construction and training quality of the reward model itself, rather than the full policy optimization pipeline. This is fundamentally different from value-based networks, which are only one component within a policy training loop and cannot be meaningfully evaluated in isolation. In contrast, a reward model can be quantitatively evaluated independently of downstream policy optimization. We fully agree with your suggestion, and integrating Search-Gen-V into a complete RL training framework to assess its end-to-end benefits is an excellent direction for future work.
>
>
> ## Weakness 4
>
> Thank you for your suggestion. We acknowledge that a systematic study of rubrics is indeed important. In our current experiments, we performed multiple spot checks and found that the rubrics—after undergoing several filtering steps, including relevance filtering, timeliness filtering, and deduplication/merging—met our expectations. Therefore, we did not conduct more fine-grained experiments in this work. We fully agree with your point and plan, in future work, to further explore methods for constructing higher-quality rubrics and to carry out a detailed analysis of rubric quality.

---

> ### Author Response · Authors · 2025-11-26
> **Response to Reviewer oyh1 (part #2)**
>
> ## Question 1
>
> For a single query, the total time required to mine rubrics averages around 1–2 hours, as it involves performing BFS over a web-scale corpus of segments until convergence—i.e., until no new segments are judged to be useful. Although this may seem time-consuming, there are several mitigating factors:
>
> 1. The time depends on the complexity of the query; simpler queries can be completed in just tens of minutes.
> 2. Our implementation uses concurrent mining, which relies on the resources of the corpus retrieval server and the LLM-as-Judge, so the actual wall-clock time is not equivalent to serial runtime.
>
> In our experiments, we employed 100 workers, and under continuous operation for 24 hours, approximately 1.9k rubrics can be mined. This allows us to quickly accumulate a sufficient volume for training purposes.
>
> ## Question 2
>
> Currently, we employ a multi-step filtering process combined with manual spot checks to remove low-quality rubrics. As you pointed out in Weakness 4, a systematic study of rubric quality is indeed necessary. We plan to further investigate issues related to rubric quality in future work.
>
> ## Question 3
>
> We currently use a two-level weighting scheme, with vital (1.0) and okay (0.5). This design stems from concerns about the reliability of the LLM-as-Judge; allowing it to generate weights automatically might introduce errors due to inaccurate judgments. Therefore, we adopt a simple weighted aggregation, as described in Equation (3) of our paper. In future work, we plan to explore more generalized and dynamic aggregation methods. This will be combined with the detailed analysis of rubric quality, including the assessment of rubric weights. Thank you for raising this insightful question.
>
> ---
>
> Once again, we sincerely thank you for your insightful review and valuable suggestions. We hope that our responses have addressed some of your concerns and look forward to further discussions.
>
> ---
>
> [1] Malik, S. et al. “RewardBench 2: Advancing Reward Model Evaluation.”
>
> [2] Saad-Falcon, J. et al. “LMUnit: Fine-grained Evaluation with Natural Language Unit Tests.”
>
> [3] Liu, C. Y. et al. “Skywork-Reward-V2: Scaling Preference Data Curation via Human-AI Synergy.”

---

### Official Review · Reviewer_aboG · 2025-10-30

**Soundness:** 2
**Presentation:** 1
**Contribution:** 1
**Rating:** 2
**Confidence:** 4

**Summary:**

The paper proposes a **nugget-as-rubric** reward paradigm to deliver verifiable rewards for search-augmented models, train a 4B generative discriminator, Search-Gen-V, to assign verifiable scores for both short-form and long-form tasks (short/long text), which can be used as RL rewards or evaluation signals. The training follows two distillation-style stages: SFT → RL.

**Strengths:**

1. Proposes nugget-as-rubric to uniformly model short-form and long-form tasks, enabling a consistent, verifiable reward across settings.
2. Trains a 4B Search-Gen-V via two stages (SFT → RL) whose effectiveness approaches Qwen3-235B-A22B-Instruct-2507.

**Weaknesses:**

### Method

1. The key notion of **“atomic golden information points (nuggets)”** within nugget-as-rubric is not explained rigorously and lacks a precise, formal definition. If this concept is derived from prior paper, the manuscript lacks **explicit citations** to those sources.

2. In the RL training of Search-Gen-V, the format reward weight reaches 30%, which diverges from some mainstream setups (e.g., DeepSeek-Math). This might bias the model toward learning the format reward. It is recommended to provide reward curves to make the training dynamics clearer.

### Baselines

1. The evaluation datasets are limited: each of the short-form and long-form settings is validated on only 1 dataset, so generalization is not convincingly demonstrated.

2. There is a lack of comparisons with other evaluation metrics. In Figure 4, the short-form workloads are compared against EM, but for long-form tasks there is no comparison to the original metrics of DeepResearch Bench (or other long-form benchmarks).

3. Baseline coverage is insufficient. The method is mainly compared to other base models; it should also be compared to the generative reward model or the scalar reward model mentioned around line 159.

### Experiments

1. The experiments focus on the reward discriminant stage only. The paper does not demonstrate using Search-Gen-V rewards to actually train a search-augmented LLM, making it hard to validate the real effectiveness of Search-Gen-V. It is suggested to conduct RL experiments that compare Search-Gen-V against rule-based or reward model based in practice.

**Questions:**

1. line 293, the paper states that Gemini-Flash aligns better with human inspection. Why, then, is Qwen3-235B used to generate the rubrics?

2. line 333, the overlength penalty is introduced, but the manuscript lacks concrete details about how it is computed and applied. Could the authors clarify this component?

---

> ### Author Response · Authors · 2025-11-26
> **Response to Reviewer aboG (part #1)**
>
> Dear Reviewer aboG,
>
> We sincerely thank you for your valuable feedback. Below we provide our responses to weaknesses you raised.
>
> ## Weakness of Method
>
> Regarding **Weakness 1**, we sincerely appreciate your careful and rigorous review. You are correct that our paper provides an overly brief description of the definition of nuggets. More precisely, a nugget refers to a complete unit-level claim or fact. In practice, a nugget is typically a 10–20 word declarative statement that includes a specific subject, an event description, and any relevant conditions or qualifiers, which is consistent with how we define them in our prompt templates. For example, "Starbucks has been criticized for tax avoidance and failing the Fair Trade test." is a nugget corresponding to the question “Why are people boycotting Starbucks?” Our definition follows prior work such as AutoNuggetizer [1]. While we did cite the original work elsewhere, we acknowledge that we did not explicitly clarify the definition in the section you referred to. We will revise and strengthen this description in the next version. Thank you again for pointing this out.
>
> Regarding **Weakness 2**, there is a specific reason why we set the format reward to 30%. Since our task involves batch-level rubric verification, we observed that when no format reward—or only a small format reward (10%)—was applied, the model frequently produced mismatched numbers of rubrics. For instance, with a batch size of 10, the model generated 20 judgment labels. This inconsistency severely compromises the reliability of a verifier or reward model. To mitigate this issue, we increased the proportion of the format reward, while still keeping the overall design consistent with standard reward shaping principles, ensuring that the format component does not dominate.
>
> Furthermore, we have uploaded the training and validation reward curves to the [anonymous repository](https://anonymous.4open.science/r/ICLR-Rebuttal-C82D/README.md) for your reference. These curves show that the reward continues to improve and does not saturate early, indicating that the model is learning rubric-related reward signals rather than overfitting to simple formatting heuristics. In addition, the validation reward generalizes well, steadily increasing to above 0.95, which further demonstrates that the model is not merely exploiting the format reward but is genuinely learning the intended verification behavior.
>
>
> ## Weakness of Baselines
>
> Regarding **Weakness 1**, your feedback is entirely reasonable—relying on a single evaluation dataset may indeed limit the assessment of generalization. To address this concern, we have conducted an additional evaluation on TriviaQA under the short-form workload setting. We sampled 1,000 examples from its validation set and generated answers using the same search-augmented LLMs and search tools as in our main experiments. The results are presented in Table 1. The findings show that Search-Gen-V-4B also achieves performance close to Qwen3-235B on this dataset, further demonstrating the generalizability of our reward model across different short-form benchmarks.
>
> Table 1:
> | Verifier Model  | Precision | Recall | F1   |
> |-----------------|:---------:|:------:|------|
> | Qwen3-4B        |    0.74   |  0.66  | 0.69 |
> | Search-Gen-V-4B |    0.83   |  0.75  | 0.78 |
> | Qwen3-235B-A22B |    0.83   |  0.79  | 0.80 |
>
> However, in the long-form setting, the availability of suitable evaluation datasets is indeed limited. Well-known benchmarks such as GAIA and xBench-Deep Search adopt entity-based short-form answers, which are not aligned with the long-form verification task studied in our work. Currently, both DeepResearchBench and our own validation set produce long-form answers grounded in web retrieval, making them naturally suitable for evaluating rubric-based long-form verification. Therefore, expanding experiments under the long-form workload is challenging due to the scarcity of appropriate benchmarks. We kindly ask for your understanding regarding this limitation.
>
>
> Regarding **Weakness 2**, we would like to clarify that for long-form tasks, we have already compared our method against the Comprehensiveness metric proposed in DeepResearchBench (described in line 444 and visualized on the left part of Figure 4). This metric evaluates whether an answer covers the key aspects of an industry, demonstrates holistic understanding, and avoids omitting important components—objectives that closely align with our proposed nugget-as-rubrics verifiable reward. Notably, this metric also relies on LLM-as-Judge with both reference reports and defined criteria. As shown on the left side of Figure 4 in the paper, the rubric-based evaluation scores produced by Search-Gen-V-4B exhibit strong consistency with the Comprehensiveness scores generated by Gemini-2.5-Flash, which further demonstrates the reliability of our verification model.

---

> ### Author Response · Authors · 2025-11-26
> **Response to Reviewer aboG (part #2)**
>
> Regarding **Weakness 3**, in the context of search-augmented LLMs, the available reward model baselines are actually quite limited. This is mainly due to two reasons: (1) many prior works focus on short-form workloads, where rule-based reward functions such as EM are predominantly used; and (2) works that adopt generative reward models typically rely on general-purpose LLMs, which is feasible because they also target only short-form tasks and do not require a specialized reward model. In contrast, our work explores both task extension (from short-form to long-form) and efficiency, leaving few existing baselines directly comparable to our setting.
>
> Nevertheless, we expanded our comparisons by adding two reward models from RewardBench 2 [2]:
> - ContextualAI/LMUnit-qwen2.5-72b [3]
> - Skywork/Skywork-Reward-V2-Llama-3.1-8B [4]
>
> Our Search-Gen-V-4B continues to determine the chosen output using our rubric-based scoring method. We compare Search-Gen-V-4B against these two baselines on the factuality metric in RewardBench 2, which aligns more closely with the objectives of search-augmented LLMs. The results in Table 2 further demonstrate the necessity of training a dedicated reward model, especially a smaller and more efficient one, to support dense-reward RL training.
>
> Table 2:
> | Reward Model/Verifier Model            |    Type    | Factuality Score |
> |----------------------------------------|:----------:|:----------------:|
> | ContextualAI/LMUnit-qwen2.5-72b        | Generative |       87.2       |
> | Skywork/Skywork-Reward-V2-Llama-3.1-8B | Classifier |       84.6       |
> | Ours/Search-Gen-V-4B                   | Generative |       85.8       |
>
>
>
> ## Weakness of Experiments
>
> Thank you for your suggestion. In this work, we focus specifically on the construction and training quality of the reward model itself, rather than the full policy optimization pipeline. This is fundamentally different from value-based networks, which are only one component within a policy training loop and cannot be meaningfully evaluated in isolation. In contrast, a reward model can be quantitatively evaluated independently of downstream policy optimization. We fully agree with your suggestion, and integrating Search-Gen-V into a complete RL training framework to assess its end-to-end benefits is an excellent direction for future work.
>
>
> ## Questions
>
> **Q1**: The human inspection mentioned in the paper refers to comparing two alternative rubric-verification strategies for long-form outputs, rather than evaluating the rubric generation stage of our pipeline. Our goal here is to identify a reliable method that can serve as the ground-truth verification label, which is required for conducting subsequent comparative experiments. And Gemini-2.5-Flash is a better choice than Qwen3-235B with voting.
>
> **Q2**: The overlength penalty refers to the fact that, during sampling, some generated responses exceed the maximum allowed length. In many setups, these overlength samples are directly penalized. However, some of these responses may actually be reasonable, and naive penalization introduces reward noise. We follow the overlength-penalty design used in DAPO. Due to space limitations, we were unable to include the full computation details in the paper—our apologies for any inconvenience this may have caused. In detail, DAPO introduces a soft penalty region, where responses that slightly exceed the ideal length receive a gradually increasing penalty, rather than an abrupt drop. This helps reduce noise. The penalty is applied as shown in Equation 1, and is ultimately combined with our defined total reward.
>
> Equation 1:
>
> $R_{\text{length}}(y) =0, |y| \le L_{\text{max}} - L_{\text{cache}}$
>
> $R_{\text{length}}(y) =\frac{(L_{\text{max}} - L_{\text{cache}}) - |y|}{L_{\text{cache}}}, L_{\text{max}} - L_{\text{cache}}<|y|\le L_{\text{max}}$
>
> $R_{\text{length}}(y) =-1, L_{\text{max}} < |y|$
>
> ---
>
> Once again, we sincerely appreciate your time, effort, and insightful comments. We hope our responses have clarified our design choices and helped improve your overall impression of our work. We look forward to further discussion and your continued feedback!
>
>
> ---
>
> [1] Pradeep, R. et al. “Initial Nugget Evaluation Results for the TREC 2024 RAG Track with the AutoNuggetizer Framework.”
>
> [2] Malik, S. et al. “RewardBench 2: Advancing Reward Model Evaluation.”
>
> [3] Saad-Falcon, J. et al. “LMUnit: Fine-grained Evaluation with Natural Language Unit Tests.”
>
> [4] Liu, C. Y. et al. “Skywork-Reward-V2: Scaling Preference Data Curation via Human-AI Synergy.”

---

> > ### Comment · Reviewer_aboG · 2025-11-27
> >
> > The authors’ rebuttal has partially addressed my concerns, so I have decided to raise my score. However, the authors have not provided a revised PDF manuscript that comprehensively incorporates the improvements suggested by the reviewers. For example, the definitions of key concepts and the  some experimental results are still not fully clarified, and in the reward curve provided by the authors, the validation accuracy appears to be higher than the training accuracy? which seems inconsistent with common practice in deep learning. Given that this is not my primary area , whether to further increase the score beyond borderline, I intend to wait for further discussion with the other reviewers before making a final decision.

---

> > > ### Author Response · Authors · 2025-12-01
> > > **Response to Reviewer aboG**
> > >
> > > Thank you very much for your further feedback. We are glad to hear that our previous responses were helpful in addressing your concerns.
> > >
> > > As requested, we have uploaded a revised PDF manuscript, incorporating several improvements suggested by the reviewers. All changes have been clearly highlighted in red.
> > >
> > > Regarding the observation that the validation reward is higher than the training reward, we believe this mainly stems from the design of the DAPO algorithm. As shown in Equation (9) of the paper, during training, the **sample filtering** step removes samples with reward = 1 or reward = 0 in order to magnify the advantage and improve sample efficiency. As a result, the reward distribution is altered. Moreover, because Search-Gen-V performs **batch-level** inference, the probability of obtaining fully correct outputs (i.e., reward = 1) during validation is higher than fully wrong outputs (i.e., reward = 0), leading filtering to remove a larger proportion of high-reward samples during training. Therefore, it is expected that the training reward is lower than the validation reward, which aligns with the intended behavior of advantage amplification in DAPO rather than indicating overfitting or inconsistency.
> > >
> > > ---
> > >
> > > Although all prior interactions were rolled back, we sincerely appreciate your continued engagement. We hope this follow-up further addresses your remaining concerns. Thank you again for the time and effort you have dedicated to reviewing our submission.

---

### Official Review · Reviewer_qChS · 2025-10-31

**Soundness:** 2
**Presentation:** 3
**Contribution:** 2
**Rating:** 4
**Confidence:** 3

**Summary:**

This paper introduces a unified "nugget-as-rubric" framework for reward modeling in search-augmented LLMs. It proposes an automatic pipeline to build rubrics from retrieved passages and trains an efficient 4B generative verifier, Search-Gen-V. Experiments show this 4B verifier achieves accuracy comparable to a 235B teacher model at a much lower computational cost.

**Strengths:**

1. The topic of this paper is crucial. The "nugget-as-rubric" approach provides a single, verifiable formulation that works across both short-form and long-form tasks.
2. The automatic rubric construction pipeline reduces the need for costly manual annotation and helps to mitigate the "pool bias" found in traditional passage-labeling methods.
3. The 4B Search-Gen-V model is highly efficient, addressing the high computational cost of generative rewards. It maintains strong performance, closely matching a 235B teacher model's judgments after a two-stage training strategy.

**Weaknesses:**

1. The reliability of the proposed method needs further demonstration.
    1. The correctness of the automatically generated rubrics is not independently verified. The pipeline's heavy reliance on an LLM-based Judge ($\Psi$) means any bias or errors from this Judge are propagated into the "ground truth" rubrics.
    2. The "golden" verification labels are derived from a teacher model (Gemini-2.5-Flash) whose own accuracy is not rigorously validated. Although the appendix includes a small human preference comparison showing a slight advantage over Qwen, this is insufficient to establish that the teacher has adequate labeling capability. Consequently, the reported F1 scores primarily reflect the student model's high *fidelity* to a potentially flawed teacher, rather than true factual accuracy.
2. The paper lacks comparative experiments with more rule-based metrics, such as F1-score or ROUGE on short-form tasks, which are more robust baselines than Exact Match. Furthermore, the paper does not compare the reward accuracy against other powerful reward modeling approaches, nor does it include the end-to-end RL training comparison to validate the improvement over other reward modeling approaches.
3. While the research is critical, the method is only suitable for knowledge verification for search-augmented LLMs and only on one dataset.
4. The performance improvement of RL is limited. And the difference between Search-Gen-V-1.7B and Search-Gen-V-4B is 0.06 in average. Do these indicate the task is not very difficult?

**Questions:**

Please see Weakness.

---

> ### Author Response · Authors · 2025-11-26
> **Response to Reviewer qChS (part #1)**
>
> Dear Reviewer qChS,
>
> We sincerely appreciate your time and effort in reviewing our submission, and we are grateful for your recognition of the motivation behind our work. In this response, we aim to clarify several potential misunderstandings and provide additional experimental results to further address your concerns.
>
> ## Weakness 1
>
> Thank you for your careful and insightful comments. We understand that your concerns regarding reliability primarily focus on the involvement of LLM-as-Judge in our pipeline. We fully agree that this trend brings both opportunities and potential risks.
>
> Regarding **Weakness 1.1**, our rubric construction pipeline incorporates multi-stage filtering. During the web-mining stage, we filter out irrelevant pages as much as possible, and an additional low-quality filter is applied during rubric generation. Moreover, based on our cited paper - AutoNuggetizer[1], which demonstrated a high level of agreement between rubrics (nuggets) generated by LLM-as-Judge and those generated with human assistance, we gained confidence in leveraging LLM-as-Judge as a core component in our pipeline.
>
> Regarding **Weakness 1.2**, we performed manual inspection and compared multiple strategies, ultimately selecting Gemini-2.5-Flash as the teacher model. We further tested and refined the effectiveness and consistency of prompt templates. In addition, the paper of AutoNuggetizer[1] also provides empirical evidence showing strong agreement between large-scale teacher-model annotations and human annotations, with Kendall’s $\tau=0.783$, which supports the feasibility of using LLM-as-Judge for rubric verification.
>
> While we rely on prior findings and incorporate several optimization steps, we agree that making LLM-as-Judge an absolutely reliable proxy still requires continuous refinement and validation. This also reflects an inherent trade-off within the scope of our work: rubric-based verification for long-form content is significantly more challenging than entity-based verification for short-form answers. Extending from short-form to long-form introduces substantial difficulty, and even manual annotation struggles to attain true factual accuracy for long-form responses. As a result, it remains difficult to simultaneously achieve both broad task coverage and perfect factual reliability. We view this as a key direction for future work.
>
>
> ## Weakness 2
>
> Within rule-based metrics, it is true that multiple functions can be applied across various QA-style tasks. Among them, Exact Match (EM) is by far the most widely used—both as an evaluation metric and as a reward function. More importantly, non-binary metrics such as F1 or ROUGE cannot provide a definitive correct/incorrect judgment. “Partially correct” labeled answers may still be factually wrong. For example, Barack Obama vs. Michelle Obama: although they share partial lexical overlap, they refer to entirely different entities. Therefore, EM is the most representative metric for short-form tasks and yields the lowest false-positive recall rate, which is why we chose EM for comparison.
>
> In addition, we strongly agree with your suggestion that comparisons with more powerful reward-modeling approaches would further strengthen the work. However, in the context of search-augmented LLMs, the available reward model baselines are actually quite limited. This is mainly due to two reasons: (1) many prior works focus on short-form workloads, where rule-based reward functions such as EM are predominantly used; and (2) works that adopt generative reward models typically rely on general-purpose LLMs, which is feasible because they also target only short-form tasks and do not require a specialized reward model. In contrast, our work explores both task extension (from short-form to long-form) and efficiency, leaving few existing baselines directly comparable to our setting.
>
> Nevertheless, we expanded our comparisons by adding two reward models from RewardBench 2 [2]:
> - ContextualAI/LMUnit-qwen2.5-72b [3]
> - Skywork/Skywork-Reward-V2-Llama-3.1-8B [4]
> Our Search-Gen-V-4B continues to determine the chosen output using our rubric-based scoring method. We compare Search-Gen-V-4B against these two baselines on the factuality metric in RewardBench 2, which aligns more closely with the objectives of search-augmented LLMs. The results in Table 1 further demonstrate the necessity of training a dedicated reward model, especially a smaller and more efficient one, to support dense-reward RL training.
>
> Table 1:
> | Reward Model/Verifier Model    | Type    | Factuality Score |
> |----------|:--------:|:-------:|
> | ContextualAI/LMUnit-qwen2.5-72b  | Generative |   87.2 |
> | Skywork/Skywork-Reward-V2-Llama-3.1-8B | Classifier | 84.6   |
> | Ours/Search-Gen-V-4B   | Generative | 85.8  |

---

> ### Author Response · Authors · 2025-11-26
> **Response to Reviewer qChS (part #2)**
>
> Regarding applying Search-Gen-V in end-to-end RL training, we clarify that this work focuses on constructing and improving the reward model itself, rather than on a full policy optimization pipeline. This is fundamentally different from value-based networks, which are only one component within a policy training loop and cannot be meaningfully evaluated in isolation. In contrast, a reward model can be quantitatively evaluated independently of downstream policy optimization. We fully agree with your suggestion, and integrating Search-Gen-V into a complete RL training framework to assess its end-to-end benefits is an excellent direction for future work.
>
>
> ## Weakness 3
>
> Thank you for recognizing the motivation behind our research. While our broader focus is indeed on knowledge verification for search-augmented LLMs, we believe this aspect represents one of the most challenging problems in the field. Our motivation comes from two key factors:
>
> (1) Task-scale expansion. Existing verifiable reward mechanisms primarily target short-form responses, but there is an urgent need to extend them to long-form settings. This is why the generalized formulation and model we propose are necessary—they are not restricted to any single dataset and can accommodate both short-form and long-form supervision scenarios.
>
> (2) Efficiency and cost. Generative reward models are inherently expensive to deploy, and when used in practical RL training, they must also meet strict throughput requirements—especially under dense-reward paradigms that include process-level rewards. Therefore, the reward model must be as compact as possible (4B) and highly efficient (supporting batched verification) to make RL training feasible.
>
>
>
>
> ## Weakness 4
>
> Thank you for your thorough review, and we truly appreciate the opportunity to discuss this point with you. Although the performance improvement brought by RL is not large, RL significantly stabilizes the evaluation quality of Search-Gen-V. During training, we observed that after SFT, the model occasionally produced a mismatch in the number of generated rubric labels, such as outputting 20 judgment labels when the batch size was 10. This issue is critical for a robust verifier or reward model. To prevent such inconsistencies from propagating to broader use cases, we introduced the second-stage RL phase and incorporated a format-consistency reward, which successfully resolved the problem. Therefore, the improvements brought by RL are multi-dimensional and necessary.
>
> In addition, we have also reflected on deeper underlying factors. We respectfully believe that the limited gains from RL may relate to the inherently sparse reasoning patterns of small models, as also discussed in DeepSeek-R1. Consequently, the performance improvements achievable through RL for such small models are naturally constrained.
>
> ---
>
> We sincerely appreciate your time and effort in reviewing our work. We hope our clarifications address your concerns and lead to a more favorable view of our work. We look forward to further discussion!
>
> ---
>
> [1] Pradeep, R. et al. “Initial Nugget Evaluation Results for the TREC 2024 RAG Track with the AutoNuggetizer Framework.”
>
> [2] Malik, S. et al. “RewardBench 2: Advancing Reward Model Evaluation.”
>
> [3] Saad-Falcon, J. et al. “LMUnit: Fine-grained Evaluation with Natural Language Unit Tests.”
>
> [4] Liu, C. Y. et al. “Skywork-Reward-V2: Scaling Preference Data Curation via Human-AI Synergy.”

---

### Author Response · Authors · 2025-12-01
**Summary for the AC**

Dear AC,

We sincerely appreciate the significant effort you have devoted to reviewing submissions for this conference. The purpose of this comment is to provide an objective summary of the discussions during the rebuttal period, with a focus on the key weaknesses identified and the corresponding improvements we have made. We hope this summary will assist you in making the final decision on our paper.

---

## Weakness 1: Lack of comparison with preference-based reward model baselines (*Reviewer qChS, aboG, oyh1*)

Description: Reviewers suggested that our experiments lacked comparisons with powerful reward model baselines.

Our Response: The initial reason for not including such comparisons is that most widely used reward models are *preference-based*, whereas our work focuses on *verifiable* rewards. To address this concern, we incorporated **RewardBench 2** as an additional benchmark and compared our model against the top two large-scale models on the leaderboard in terms of factuality scores. The results show that Search-Gen-V-4B achieves comparable performance while using significantly fewer parameters, thereby improving efficiency.

| Reward Model/Verifier Model    | Type    | Factuality Score |
|----------|:--------:|:-------:|
| ContextualAI/LMUnit-qwen2.5-72b  | Generative |   87.2 |
| Skywork/Skywork-Reward-V2-Llama-3.1-8B | Classifier | 84.6   |
| Ours/Search-Gen-V-4B   | Generative | 85.8  |


## Weakness 2: Lack of more evaluation datasets (*Reviewer aboG, oyh1, r9Uu*)

Description: Reviewers felt that evaluating only one dataset per workload was insufficient.

Our Response: In practice, most available datasets are short-form. Aside from our constructed validation set and DeepResearch Bench, long-form datasets were scarce at the time of submission. Therefore, we expanded our short-form evaluations by adding **TriviaQA**, and conducted experiments under the same settings as HotpotQA. The results demonstrate that Search-Gen-V-4B matches the performance of Qwen3-235B while being far more parameter-efficient.

| Verifier Model  | Precision | Recall | F1   |
|-----------------|:---------:|:------:|------|
| Qwen3-4B        |    0.74   |  0.66  | 0.69 |
| Search-Gen-V-4B |    0.83   |  0.75  | 0.78 |
| Qwen3-235B-A22B |    0.83   |  0.79  | 0.80 |


## Weakness 3: Concerns about runtime (*Reviewer oyh1, r9Uu*)

Description: Reviewers expressed concerns that rubric construction appeared too time-consuming.

Our Response: First, the running time of rubric construction process depends on the difficulty of the user’s query. And in our implementation, rubric mining is executed in a highly concurrent manner rather than sequentially. The runtime depends on available GPU memory, CPU memory, and other system resources. With 100 parallel processes in our real testing, the mining speed is substantially faster than the serial estimate. Under continuous operation for 24 hours, our pipeline can mine approximately 1.9k rubrics, allowing us to rapidly accumulate sufficient data for training.

## Weakness 4: Concerns about format reward during RL (*Reviewer qChS, aboG*)

Description: Reviewers questioned the necessity of the RL stage and the use of a 30% format reward.

Our Response: The RL stage is indeed essential. After SFT, we observed occasional mismatches in the number of generated rubric labels—for example, the model sometimes produced 20 judgment labels when the batch size was 10. Such inconsistencies significantly undermine the reliability of a verifier or reward model. The second-stage RL, with a relatively higher proportion of format reward, effectively resolves this issue. Additionally, we provide *reward curves* showing that training is not misled by the format reward and that the model generalizes well on the validation set.


---


In this rebuttal, we actively responded to every question raised by each reviewer. Prior to the rollback, Reviewer aboG responded to our clarifications and provided positive feedback, increasing the score from 2 to 4. And we further replied to the additional questions raised by Reviewer aboG. Following the instructions, we also uploaded a revised version of the paper, with all changes clearly highlighted in red. We kindly hope that our interactions with the reviewers can be taken into consideration during your decision-making process. Thank you very much.


---


Your efforts have been instrumental in resolving the API security incident that arose during this conference and in maintaining the stability of our community. We would like to express our sincere respect and appreciation for your dedication.

Sincerely,

The Authors

---

### Meta-Review · Area_Chair_55Dk · 2026-01-04

**Summary:**

This paper proposes a "nugget-as-rubric" framework for verifiable reward construction in search-augmented LLMs, covering both short-form and long-form workloads. It introduces an automated rubric construction pipeline and trains a compact 4B-parameter verifier via distillation and RL. Reviewers found the idea intuitive and potentially useful, but raised concerns about its reliability, novelty, and the strength of validation.

**Reviewer Concerns:**

Reviewers consistently raised several issues. First, the reliability of automatically generated rubrics remains not sufficiently validated, as evaluation largely measures agreement with teacher LLMs and lacks independent verification at scale. Second, the contribution is viewed as incremental (oyh1), which combines existing ideas in nugget extraction, rubric-based verification, and reward distillation. This is fine if the evaluation is thorough and solid. However, it feels that that the evaluation breadth is somewhat limited, with each workload demonstrated on a small number of datasets and without convincing e2e RL policy improvement.

The rebuttal addressed some concerns with additional comparisons (e.g., RewardBench), added clarifications on rubric construction, added reward curves, and expanded short-form evaluation. However, these additions only partially resolve the main concerns around reliability, novelty, and empirical evaluation.

**Reviewer Scores:**

Reviewer qChS: 4, borderline. Likely did not change the score.

Reviewer aboG: Increased from 2 to 4 after rebuttal. Unfortunately, the discussion on "validation reward higher than training reward" halted due to the incident. Although the authors provided an explanation from the design consideration of DAPO, it is not entirely clear if it is sufficient to catapult the paper from 4 to 6.

Reviewer oyh1: 6, marginally positive but noted multiple limitations.

Reviewer r9Uu: 6, viewed the idea as useful but incremental and empirically limited.

---

### Decision · Program_Chairs · 2026-01-26

Reject